# Spatiotemporal patterns and prediction of landscape ecological security in Xishuangbanna from 1996–2030

**Zhuoya Zhang, Hailong Ge, Xiaona Li, Xiaoyuan Huang** ⓘ **\*, Siling Ma, Qinfei Bai**

Faculty of Geography and Ecotourism, Southwest Forestry University, Kunming, China

\* hxy19792721@163.com

**Data Availability Statement:** Our base data is Landsat 8 remotely sensed imagery data, obtained from the Geospatial Data Cloud (http://www.gscloud.cn/), which can be accessed directly by visiting the site, enter the Geospatial Data Cloud

## Abstract

In recent years, the landscape ecological security of Xishuangbanna in southwest China has become an essential factor affecting the cross-border ecological security in South Asia and Southeast Asia. Based on the change of land use in Xishuangbanna, with the help of "3S" technology, landscape ecology theory, and gray prediction model, the spatial and developmental trends of landscape ecological security in Xishuangbanna from 1996–2030 could be determined. In more than 20 years, the woodland landscape area in Xishuangbanna decreased, and the fragmentation of construction land has increased overall. In 1996, the overall landscape ecological safety was good, with 63.5% of the total area of grade I and II. In 2003, the proportion of the grade I and grade II areas decreased, with landscape ecological security problems appearing. In 2010, the overall landscape ecological security area reached 74.5%, the largest proportion in more than 20 years. The grade V area accounted for only 9% and was mainly distributed on the border of Menghai County and central Jinghong City. In 2017, The grade IV and V areas was further increased, and the ecological security problem intensified. The prediction results showed that from 2023 to 2030, the regions of grades I and II increased, but the proportion of level V regions increased. Furthermore, the grade IV transformed to grade V rapidly, reaching its highest value in more than 20 years. From 1996 to 2030, the landscape ecological security space significantly evolved, showing an evident "east-south" trend in movement and eventually shifting to the southeast.

## Introduction

Global socio-economic development has led to high-intensity land use and rapid land-use changes, putting great pressure on the natural environment [1]. The ecological crisis and disasters caused by the unsustainable utilization of natural resources have a direct impact on the regional landscape pattern, sustainable development, economic competition, and national security in the context of globalization. Achieving ecological security has garnered significant attention from governments worldwide due to the mounting pressure caused by the sharp changes in the global environment, which directly impact the survival and development of

website, click on Advanced Search, select Landsat 8 OLI_TIRS Satellite Digital Product, and select the appropriate spatial location, time range and cloud amount for searching, with special access or request privileges for those without privileges or request privileges.

**Funding:** This study was supported by National Natural Science Foundation of China in the form of grants awarded to: (Grant 42171240), XL (Grant 31901322), ZZ (Grant 42377444), Research start-up fund from Southwest Forestry University in the form of a grant awarded to ZZ (Grant 112125) and Yunnan Dianchi Lake Protection and Management Foundation, Biodiversity Survey in Dianchi Lake Basin in the form of a grant awarded to ZZ (Grant 2063010). The specific roles of these authors are articulated in the 'author contributions' section. The funders had no role in study design, data collection and analysis, decision to publish, or preparation of the manuscript.

**Competing interests:** The authors have declared that no competing interests exist.

humanity. Research on the topics of sustainable economic and social development has become a global hot topic [2–7]. The International Institute of Applied Systems Analysis (IASA) first proposed the concept of ecological security in 1989 [8], referring to the possibility of the destruction or evolutionary trends of the ecological environment that are necessary to maintain the sustainable development of human society, as well as the potential impact of each trend on the human ecological security space [9, 10].

Landscape not only consists of natural and social resources, but it is also an object of human economic development. Human economic development activities primarily occur at the landscape level, making it an appropriate scale to study the environmental impact of human activities [11–13]. The principles and methods of landscape ecology can help preserve the benefits of the component features of landscape structure. This is because the landscape, region, and watershed are interconnected on an organizational scale and are relatively small. These characteristics are conducive to maintaining the accuracy of the research results on the scale, providing a better scientific understanding, and explaining the essence of ecological security at both large and medium scales [11, 12]. The evolution of a landscape pattern leads to changes in the spatial structure of the landscape, which is intuitively reflected in the changes in the ecosystem structure and composition, ultimately affecting ecological security [13, 14]. The landscape scale primarily examines the rationality and smoothness of the landscape structure, while pattern and process research provides crucial information for ecological protection and management [13]. The interference and stress on the landscape pattern have gradually increased, making landscape ecological safety an increasingly hot topic in the field of "landscape pattern-ecological process" [15–17].

Along with rapid shifts in land development and utilization, as well as environmental deterioration, these factors directly impact the regional landscape, sustainable development, and ecological security [18–20]. For the potential ecological security risks caused by human factors, the process of land-use change plays a decisive role in regional ecological security [21, 22]. Therefore, analyzing the research on the current process and trend of land-use changes and integrating it with ecological security is beneficial in providing a scientific basis for formulating land-use planning under ecological safety conditions. The landscape is the basic unit for managing regional ecological environments. Human development activities on land have a significant impact on the landscape, leading to drastic changes in its structure and function [23]. The research and evaluation of ecological security, based on landscape structure, are conducive to the combination of regional ecological security status quo and dynamic research. At present, many studys have been conducted on the relationship between landscape pattern indices, landscape ecological security, and assessment methods [24–26]. Previous studies have largely focused on (1) understanding the mechanisms driving landscape ecological security, with a particular focus on the multi-application pressure-state-response (PSR) model and spatial heterogeneity, selecting watersheds, plains, wetlands, cities, and other areas for research [27–29]; (2) conducting scenario simulations and future trend predictions, using methods like CA-Markov to predict land use types and analyze their future risk status [30]; (3) Evaluate the cumulative effects of risk factors in the landscape, using land use data and landscape indices that directly reflect ecological risks in the structure and composition of the landscape pattern, based on the theory of process correlation [31, 32]. The evaluation methods are becoming increasingly accurate with the expanding research, but a unified standard system has not yet been formed. At present, research on landscape ecological security is relatively weak and uses Grid GIS.

Economic globalization has rapidly infiltrated the influence of resources and environmental factors into all levels of national security, international economy, and trade. With the strengthening of geopolitical and economic cooperation between China and neighboring countries,

cross-border ecological security has become an important part of national ecological security [33]. Xishuangbanna is located in the southwest of China, with most of its administrative area belonging to the longitudinal ridge valley. It is a key area reflecting the evolutionary events of Earth. The border is 966 km long. The Lancang-Mekong River Basin from Xishuangbanna Mengla County, extends across Myanmar, Laos, Thailand, Cambodia, and Vietnam, making it an important cross-border basin. Its upstream hydropower development and agricultural production were previously controversial. Landscape ecological security caused by environmental changes in international river basins and border land use and development in Xishuangbanna may become a global ecological problem. In addition, Xishuangbanna is located on the northern edge of the tropical rainforest that provides a suitable habitat and vertical distribution of vegetation. The forest vegetation presents the widest, most complex, and most complete structure, and it holds an ecological system with the highest biological production, providing a variety of ecological types and a key national cultural area [30]. Xishuangbanna has been supporting China's economic construction as an export base, such as through forests and minerals. Its rapid changes in the ecosystem and species diversity are affected by the dual interaction of natural evolution and human activities. Although the forest coverage rate is high, the natural forest damage is serious, especially for the large proportion of economic forests (rubber forest) and economic plantations (tea), resulting in a fragile ecological environment in the region, which will lead to greater ecological pressure. Existing research results on land use [34], landscape pattern [10], and vegetation [35] reveal the ecological security problems of Xishuangbanna, but most of them are qualitative analysis of a single type of ecological security risk. There are still few studies on the construction of landscape security through the landscape risk index, and it is difficult to comprehensively and systematically evaluate the ecological security status of Xishuangbanna.

In addition, there are still few studies on constructing landscape security through a landscape risk index, and it is difficult to comprehensively and systematically assess the ecological security status of Xishuangbanna.

The present study of the Xishuangbanna area was conducted using 1996 to 2017 remote sensing image data collected through ENVI and ArcGIS software analysis that analyzed Xishuangbanna land use / land cover change. This data was combined with the two phases of a forestry survey data comparative analysis and field investigation proofreading verification, which was then examined with the GIS grid method and landscape pattern index method. The research area, land use landscape pattern, ecological security space, and space pattern provided a complete in-depth study. The Xishuangbanna ecosystem has been disturbed on an unprecedented scale, and the ecological security has become prominent, attracting extensive international attention. This study is essential for China to build an ecological system that guarantees cross-border security and the security of important resources, solve key scientific problems in ecological and infrastructure construction in the mountainous areas in southwest China, promote protective and developmental coordination, and to play a crucial role in cross-border ecological security research and sustainable development.

## Study area and study methods

### Overview of the study area

Xishuangbanna Dai Autonomous Prefecture is located on the southwest border on the southern edge of Yunnan province (Fig 1). With a border that is 966 km long, it is connected to Laos, Burma, Vietnam, and Thailand, and it is an important channel for South Asia and Southeast Asia [36]. In addition to being the location of China's largest preserved area of tropical rainforest, it also possesses the most tropical forest ecosystem types in China. Its vegetation types include tropical seasonal rainforest, montane rainforest, subtropical evergreen broad-

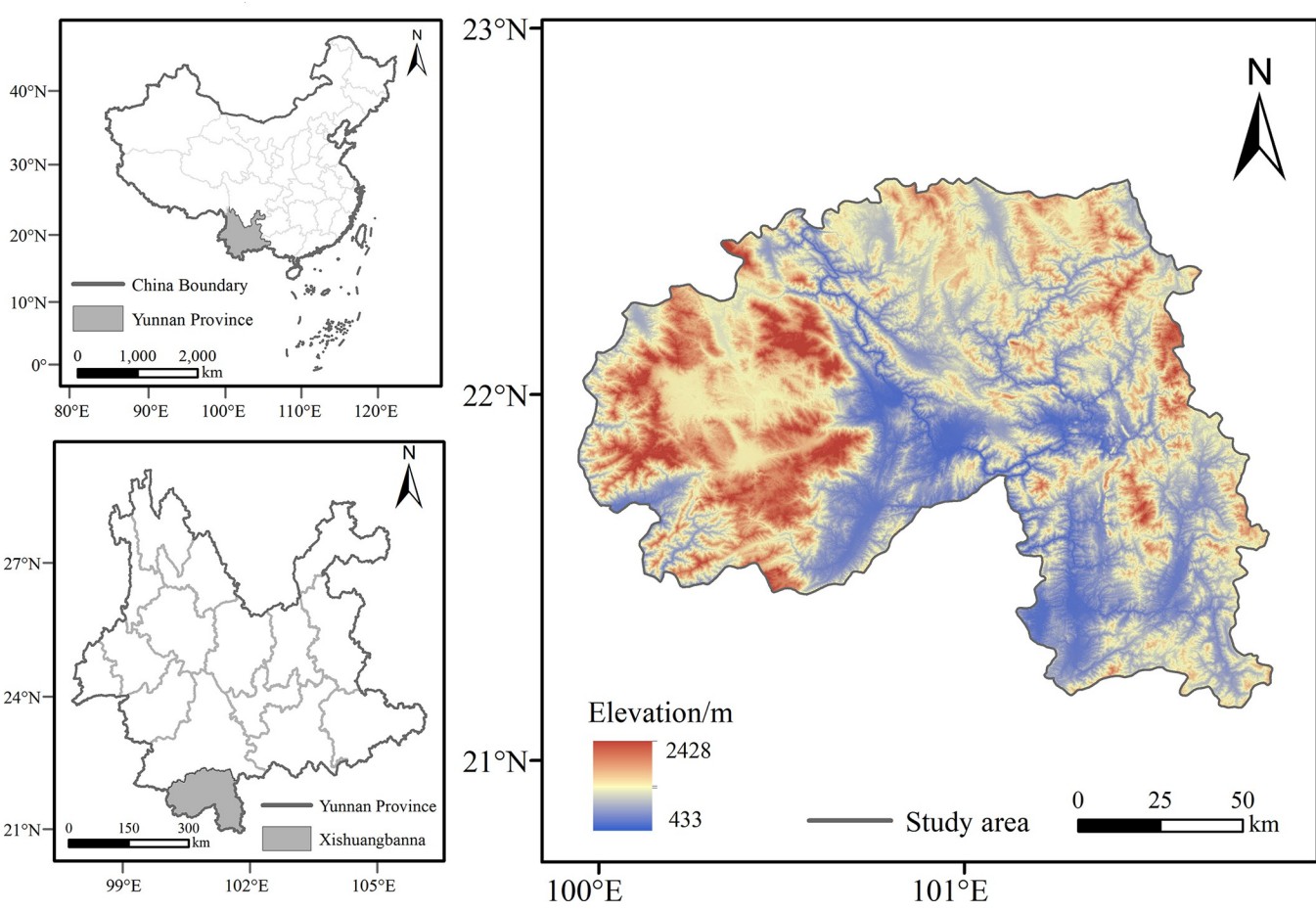

**Fig 1. Location of the study area.**

leaved forest, deciduous broad-leaved forest, and warm coniferous forest. Since the 1950s, local farmers in Xishuangbanna began to grow large rubber forests because of the rapid economic development and industrial demand for rubber products. Rubber forest planting then underwent a period of rapid development, with rubber forest expansion causing a large area of the local original tropical rainforest due to experience logging and burning [37]. Studies have shown that land use / land cover changed significantly in Xishuangbanna from 1965 to 2007. During this time, rubber gardens, dry land, and tea gardens increased, and rubber gardens became the most important land-use type, with its main transformation sources being intermittent land, woodland, and shrub forest [38, 39]. Because rubber plantations and tea gardens need to be planted under specific natural conditions, the planting area is more concentrated in space, causing increased landscape vulnerability. The replacement of natural forests into artificial forests not only changed the structure of the landscape type, but also caused changes in the ecological functions of each landscape type. The sharp changes in Xishuangbanna land use / land cover pose great risks to the local and cross-border ecology of Xishuangbanna.

## Data source and processing

The data used in this study mainly includes Landsat series remote sensing images, Digital Elevation Model (DEM) data (30 m resolution), Xishuangbanna II survey data from 2006 and

2017, and an administrative vector boundary map. For the remote sensing data, four images were selected from March to April in the years 1996, 2003, 2010, and 2017. These images had cloud volumes ranging from 0 to 2%. Remote sensing images and DEM data were downloaded from the Geospatial Data Cloud (http://www.gscloud.cn/), and the data projection is UTM WGS-84. In 2006 and 2017, the Xishuangbanna Forest Resources Class II survey data and administrative vector boundary map were derived from the Xishuangbanna Forestry Survey and Planning Institute.

ENVI 5.1 software was used, and remote sensing images in the study area were obtained through image cutting and splicing. According to the national standard of Classification of Land Use (GB / T21010-2017), combined with the current situation of land use in Xishuangbanna, particularly the large-area planting of rubber forests and tea gardens, the land use types in the research area are divided into forest land, rubber forests, tea gardens, cultivated land, construction land, and water areas. Based on field investigations, as well as spectral information from remote sensing data, forest resources class survey data and Google Earth related data from 2006 and 2017 were referenced to build the Xishuangbanna land use remote sensing interpretation mark. By using the support vector machine (SVM) supervision classification method of Xishuangbanna land-use type data interpretation, the research area five land-use types was obtained (Fig 2). With the accuracy of the GPS distribution data from the field survey conducted in August 2018 and the two-phase Class II survey data with the confusion matrix, 100 random samples were drawn from each phase plot for validation. In this study, a total of 305 GPS sampling points were distributed along the main roads in Xishuangbanna Prefecture. Among these points, there were 68 rubber plantation areas, 73 tea gardens, 85 forested areas, 23 developed lands, 16 water bodies, and 40 cultivated lands The total accuracy Kappa coefficients after phase 5 image classification were 85.9%, 86.7%, 89.9%, 93.5%, and 87.3%, respectively, which met the study needs.

## Study methods

By analyzing the remote sensing image data of Xishuangbanna for more than 20 years, comprehensive landscape indices and landscape ecological security were constructed, and the regional ecological security degree of Xishuangbanna was then analyzed and evaluated by the spatial analysis methods.

**Construction of comprehensive landscape index.** Affected by artificial disturbance, the Xishuangbanna forest develops towards increased heterogeneity and complexity, leading to the degradation and destruction of the landscape structure, which will greatly impact the ecological environment and overall safety of the entire region. In this study, three measures of landscape fragmentation, landscape separation, and landscape advantage reflect the structural characteristics after landscape interference, and the landscape disturbance index, vulnerability index, and loss index were selected to measure the causality of the landscape ecological environment [40]. The index of each landscape was calculated using Fragstats 4.2.

1. Landscape disturbance degree index ($E_i$)
   $E_i$ reflects the extent of ecosystem interference with the different landscapes.

$$E_i = \alpha C_i + bS_i + cD_I, \tag{1}$$

Where $C_i$ is the landscape fragmentation index; $S_i$ is the landscape separation index; $D_i$ is the landscape dimension index; $a$, $b$, and $c$ are the corresponding index weights; and a + b + c = 1, according to the relevant study [11, 40]. The study area fragmentation index is the most important, followed by the separation index of the $a$, $b$, and $c$ indexes 0.5, 0.3, and 0.2.

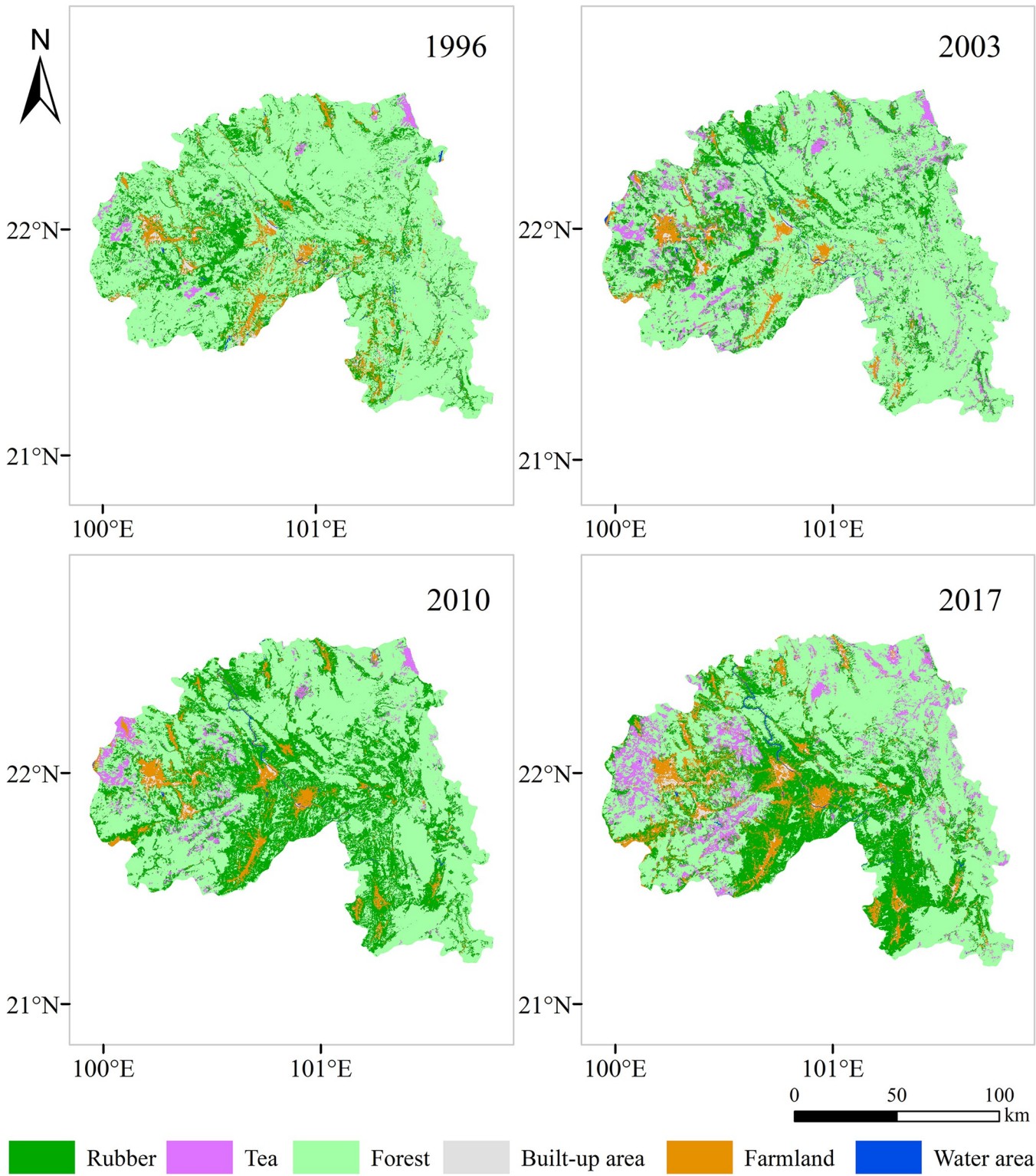

**Fig 2. Land use type changes in Xishuangbanna from 1996 to 2017.**

2. Landscape vulnerability index ($F_i$)

$F_i$ refers to the vulnerability of ecosystems to severe disturbances outside humans [14]. In this study, landscape vulnerability reflects the changing state of the landscape after the Xishuangbanna landscape type is disturbed, which is closely related to the natural succession process of landscape structure, the functional integrity, and the nature and intensity of external interference [41].

$$F_i = \alpha C_i + \beta S_i + \gamma FD_i + \delta VC_i + \theta SI, \tag{2}$$

Where $\alpha$ is the thefragmentation degree, $\beta$ is the separation degree, $\gamma$ is the dimension reciprocal, $\delta$ is the vegetation coverage index, and $\theta$ is the soil erosion index. The factor analysis describes the connection between multiple metrics with several factors and reflects most of information about the objective function with fewer factors [40]. In this paper, the factor analysis method is used to calculate the weight of the landscape vulnerability factor. Using the expert scoring method to indicate vulnerability, the construction land is 0.05, forest land is 0.10, cultivated land is 0.14, water area is 0.19, tea garden is 0.24, and rubber is 0.28.

3. Landscape loss degree index ($R_i$)

$R_i$ refers to the difference in the natural loss of the internal landscape during its external interference.

$$R_i = E_i . F_i \tag{3}$$

**Construction of landscape ecological safety degree.** Landscape ecological security refers to the response of human activities and natural stress to ecological security on a landscape scale. Spatial statistics studies the spatial distribution characteristics and spatiotemporal variation law of landscape ecological security.

(1) Determination of the evaluation unit of the study area

Through the Fishnet Analysis tool in ArcGIS 10.8, taking into account the study area's extent and the patch areas of different landscape types, following the 2–5 times principle based on patch size [42–44], the study area is divided into 243 square grid cells of 10 km×10 km each using an equidistant method. A risk index is then constructed to determine the risk value at the center point of each landscape unit, which is utilized to calculate the landscape ecological security index (Fig 3).

(2) Calculation of the landscape ecological security index

Landscape ecological security and landscape ecological risk are inverse functions [45]. Landscape Ecological Risk Index ($ERI_{ki}$):

$$ERI_{ki} = \sum_{i=1}^{n} \frac{A_{ki}}{A_k} \times R_i, \tag{4}$$

Where $A_{ki}$ is the $i$ landscape area of the $k$-th risk community, $A_k$ is the total area of the $k$-th risk community, and $R_I$ is the ecological loss index of the class $i$ landscape.

**Landscape ecological security analysis.** In this study, the spatial analysis of regional ecological safety using variation function $\gamma(h)$ in geostatistics was performed by half-variance

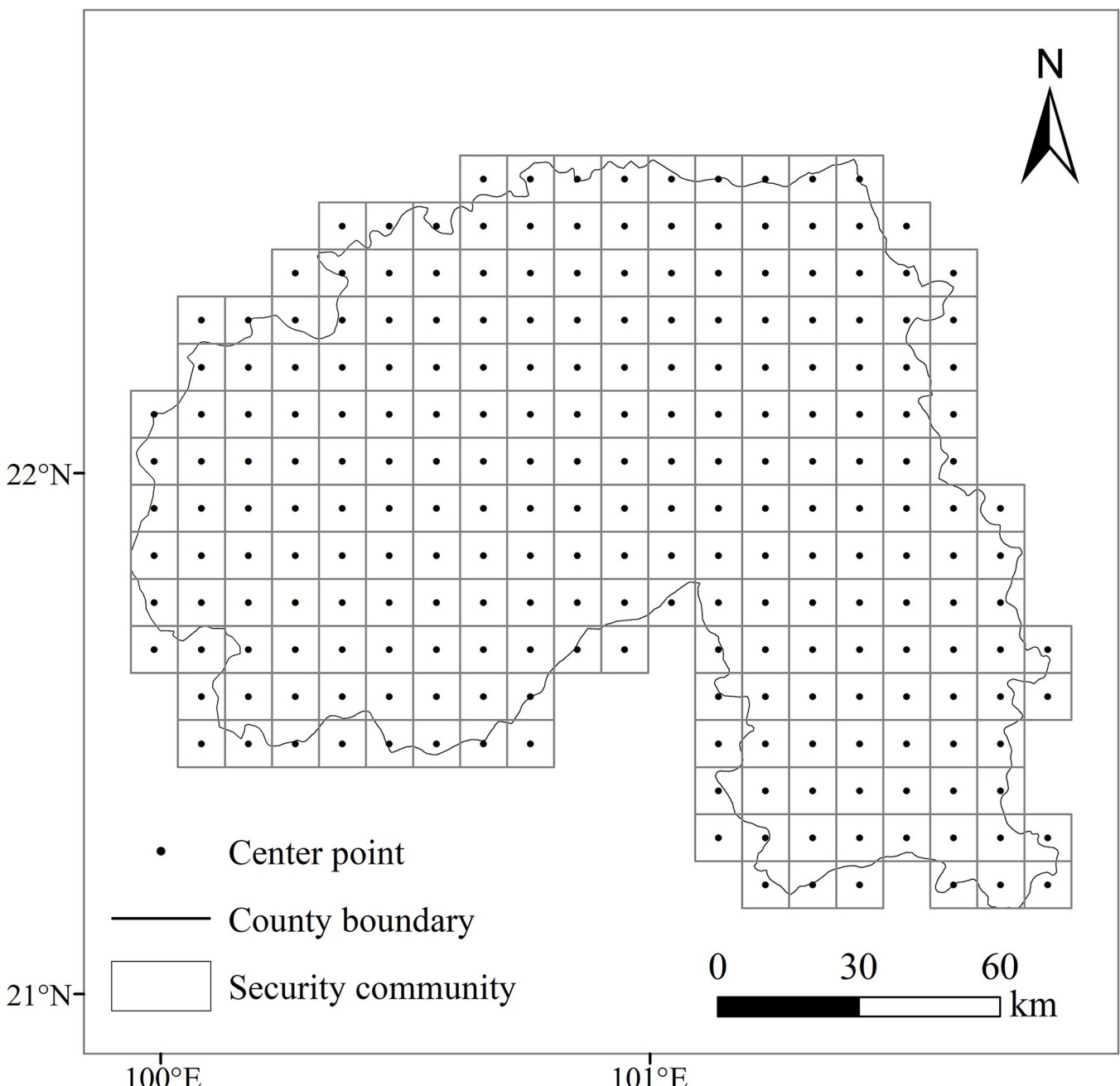

**Fig 3. The ecological security community of the study area.**

function [46]. The formula is shown as follows:

$$\gamma(h) = \frac{1}{2N(h)} \sum_{i=1}^{N(h)} [Z(x_i) - Z(x_i + h)]^2,  \tag{5}$$

Where $h$ is the step length, $N(h)$ is the interval distance is the time sample point log, $Z(x_i)$ and $Z(x_i+h)$ are the observed values of the landscape ecological security index at spatial positions $xi$ and $xi + h$, respectively.

**Spatial autocorrelation analysis.** Spatial autocorrelation analysis is divided into global spatial autocorrelation and local spatial autocorrelation, which is an algorithm used to detect the correlation between the same eigenvalues in different spatial units [42]. Moran index (*Moran's I*) is used to measure the global spatial autocorrelation of ecological risks. At the significance level, *Moran's I* takes values ranging from -1 to 1. In this paper, the GeoDa 1.2 software was used to measure the overall spatial pattern of landscape ecological security by calculating the *Moran's I* index. *Moran's I* can be calculated using the following formula:

$$I = \frac{\sum_{i=1}^{n} \sum_{i=1}^{m} w_{ij}(x_i - x)(x_j - x)}{S^2 \sum_{i=1}^{n} \sum_{i=1}^{m} W_{ij}} \tag{6}$$

$$S^2 = \frac{1}{n} \sum_{i=1}^{n} (x_i - x)^2 \tag{7}$$

Where *I* is *Moran's I* index, *n* is the total number of grids, *i* and *j* are grid *i* and grid *j*, respectively, $w_{ij}$ is the value of spatial weight matrix elements, and $x_i$ and $x_j$ are the eigenvalues of grid *i* and grid *j*, respectively. $I > 0$ indicates that the ecological risk values tend to exhibit spatial clustering (high values clustered together). $I < 0$ indicates that the ecological risk values of the landscape are dispersed (high and low values clustered together). $I = 0$ indicates that the ecological risk values of the landscape are randomly distributed in space [47]. However, since global spatial autocorrelation considers the entire study area, spatial outliers cannot be detected. Therefore, local autocorrelation was used to analyze the clustering and distribution of landscape ecological safety values in each grid and its adjacent grids:

$$V = \frac{x_i - x}{S^2} \sum_{j=1}^{n} (x_j - x) \tag{8}$$

where *V* is the local *Moran's I* for each grid and the rest of the terms are the same as in Eqs (6) and (7).

**Change of landscape ecological security focus.** The migration of the landscape center of gravity is used to reflect the changing trends and directions of regional landscape ecological conditions [48].

$$x = \sum_{i=1}^{n} (c_i \times x_i) / \sum_{i=1}^{n} c_i \qquad y = \sum_{i=1}^{n} (c_i \times y_i) / \sum_{i=1}^{n} c_i, \tag{9}$$

Where *x* and *y* are the latitude and longitude coordinates of the *i*-th landscape ecological security level, $c_i$; $x_i$ and $y_i$ are the coordinates of the *i*th patch of a landscape ecological security level. According to Formula (5), the gravity coordinates of landscape ecological security can be calculated from 1996 to 2017.

The center of gravity transfer distance and movement direction model [12] was introduced to measure the distance and direction of the center of gravity transfer at all levels of landscape ecological security.

The center of gravity of a landscape ecological security level in years *m* and *n* is *Pm* ($X_m$, $Y_m$), $P_n$ ($X_n$, $Y_n$), a surname $d_{n-m}$. The spatial distance (km) of the center of gravity transfer is from *m* to *n* years: *m* is the start year, *n* is the end year, and *c* is a constant (c = 111.11), indicating the coefficient converted from the Earth latitude and longitude coordinate unit (°) to the plane distance (km). The formula is provided as follows:

$$d_{n-m} = c \times \sqrt{(X_n - X_m)^2 + (Y_n - Y_m)^2}. \tag{10}$$

Let $\theta_{n-m}$ be the angle of gravity shift of a certain landscape ecological security level type from $m$ to $n$ years, a = 0, 1, 2.

$$\theta_{n-m} = \frac{\alpha\pi}{2} + \text{arctg}\left(\frac{Y_n - Y_m}{X_n - X_m}\right). \tag{11}$$

## Landscape ecological security trend forecast

The grey prediction model was used to predict the trends in ecological security in the Xishuangbanna landscape. By identifying the degree of development trend among system factors, generating the original data is used to find the law of system changes, generate a strong regular datGM sequence, and establish the corresponding differential equation model to predict the future trend of Xishuangbanna's landscape ecological security [15, 49].

The gray prediction model (1,1) is:

$$x^{(0)} = \{x^{(0)}(1),\ x^{(0)}(2),\ \ldots x^{(0)}(m)\} \tag{12}$$

Define the original data sequence, weaken the randomness, show its regularity, and then obtain a cumulative sequence.

$$X^{(1)} = \{x^{(1)}(1),\ x^{(1)}(2),\ \ldots x^{(1)}(m)\} \tag{13}$$

$x^{(1)}$ The derivative defined is shown as follows:

$$d(k) = x^{(0)}(k) = x^{(1)}(k) - x^{(1)}(k-1). \tag{14}$$

$z^{(1)}(k)x^{(1)}$ Make neighbors of the columns, i.e.,

$$z^{(1)}(k) = ax^{(1)}(k) + (1-a)x^{(1)}. \tag{15}$$

Thus, the differential equation of GM (1,1) is:

$$D(k) + az^{(1)}(k) = b. \tag{16}$$

k = 1,2,3,. . .,m, The differential equation of GM (1,1) is obtained to achieve the predicted value:

$$x^{(1)}(t+1) = (x^{(0)}(1) - b/a)e^{-at} + b/a. \tag{17}$$

## Results

### Analysis of comprehensive landscape index

Based on the software, the landscape pattern index of the six landscapes from 1996 to 2017 was Fragstats 4.2 and Excel (Table 1). The results show that from the dynamic change and fragmentation degree of the landscape type area, the natural woodland area decreased by $40.8\times10^4$ $\text{hm}^2$ due to the interference and influence of human factors. The fragmentation degree of natural forest landscape increased significantly, and the fragmentation of the natural forest is more dispersed.

The overall area of rubber forest land and tea gardens was on the rise, and the area of rubber forest land increased by $31.3\times10^4$ hm2 from 1996 to 2010. Compared with 1996, the rapid rise in rubber prices has prompted rubber farmers to increase the planting of rubber forests. Rubber forests decreased slowly from 2010 to 2017.

Tea gardens were added by $11.19\times10^4$ hm2 from 1996 to 2017. It is a threefold increase from 1996. Due to the continuously large planting area, the landscape area of rubber forests

**Table 1. Index of landscape patterns of various landscapes.**

| Landscape type | Area/$10^4$ hm$^2$ | Year | $C_i$ | $S_i$ | $D_i$ | $E_i$ | $F_i$ | $R_i$ |
|---|---|---|---|---|---|---|---|---|
| Forestry | 148.6 | 1996 | 0.004 | 0.037 | 1.252 | 0.263 | 0.10 | 0.026 |
| | 138.4 | 2003 | 0.004 | 0.039 | 1.250 | 0.264 | 0.10 | 0.026 |
| | 111.5 | 2010 | 0.009 | 0.063 | 0.939 | 0.211 | 0.10 | 0.021 |
| | 105.5 | 2017 | 0.008 | 0.060 | 1.268 | 0.275 | 0.10 | 0.021 |
| Rubber | 30.2 | 1996 | 0.047 | 0.258 | 1.388 | 0.379 | 0.28 | 0.106 |
| | 34.6 | 2003 | 0.049 | 0.268 | 1.375 | 0.380 | 0.28 | 0.106 |
| | 61.5 | 2010 | 0.015 | 0.108 | 0.990 | 0.238 | 0.28 | 0.067 |
| | 61.44 | 2017 | 0.020 | 0.135 | 1.323 | 0.315 | 0.28 | 0.088 |
| Tea garden | 5.01 | 1996 | 0.205 | 0.784 | 1.329 | 0.903 | 0.24 | 0.217 |
| | 11.2 | 2003 | 0.123 | 0.661 | 1.403 | 0.540 | 0.24 | 0.130 |
| | 11.3 | 2010 | 0.112 | 0.631 | 1.330 | 0.511 | 0.24 | 0.123 |
| | 16.2 | 2017 | 0.064 | 0.436 | 1.358 | 0.434 | 0.24 | 0.104 |
| | 11.3 | 2019 | 0.115 | 0.899 | 1.0593 | 0.539 | 0.24 | 0.129 |
| Cultivated field | 9.3 | 1996 | 0.115 | 0.768 | 1.376 | 0.563 | 0.14 | 0.079 |
| | 9.0 | 2003 | 0.072 | 0.724 | 1.285 | 0.510 | 0.14 | 0.072 |
| | 9.0 | 2010 | 0.049 | 0.504 | 1.286 | 0.433 | 0.14 | 0.061 |
| | 9.6 | 2017 | 0.064 | 0.456 | 1.352 | 0.439 | 0.14 | 0.062 |
| Built-up | 2.1 | 1996 | 0.082 | 1.387 | 1.253 | 0.708 | 0.05 | 0.035 |
| | 2.0 | 2003 | 0.169 | 2.030 | 1.299 | 0.953 | 0.05 | 0.048 |
| | 1.8 | 2010 | 0.085 | 2.271 | 1.159 | 0.956 | 0.05 | 0.048 |
| | 2.4 | 2017 | 0.146 | 1.723 | 1.311 | 0.852 | 0.05 | 0.043 |
| Water | 0.7 | 1996 | 0.156 | 2.712 | 1.251 | 1.142 | 0.19 | 0.217 |
| | 0.7 | 2003 | 0.622 | 2.715 | 1.339 | 1.393 | 0.19 | 0.265 |
| | 0.7 | 2010 | 0.093 | 2.769 | 1.188 | 1.115 | 0.19 | 0.212 |
| | 0.7 | 2017 | 0.111 | 2.502 | 1.222 | 1.051 | 0.19 | 0.200 |

$C_i$ is Degree of fragmentation; $S_i$ is Resolution; $D_i$ is Fractal dimension; $E_i$ is Interference degree; $F_i$ is Frailty; $R_i$ is Loss degree.

and tea gardens expands, the degree of the contiguous area is gradually enhanced, and the degree of fragmentation is decreased.

The landscape type area of the construction land shows an overall increase. From 1996 to 2010, the fragmentation of cultivated land decreased due to the extensive reclamation for farming land. Then, from 2010 to 2017, due to the acceleration of urbanization and economic and social development, part of the cultivated land was divided, and the fragmentation of the cultivated land landscape increased. On the whole, land-use change in Xishuangbanna became increasingly fierce. The landscape pattern has changed from the historically dominant natural tropical forests to large areas of artificial rubber forest and tea gardens, and the forestry land accelerated to the direction of agricultural use.

From the perspective of landscape separation, the change of landscape separation and fragmentation of the landscape of the natural forest landscape gradually increased from 0.037 to 0.060. It can be seen that the extent of the landscape of the natural forest, the separation of the landscape area of the research area increased, and the separation of cultivated land and construction land landscape increased. From the perspective of landscape dimension, the dimension of natural forest landscape increased, especially for the development of rubber forests and tea garden, which gradually decreased the shape complexity. Furthermore, the dimension index of the rubber forest, tea garden, farming land, construction land, and water landscape types are basically the same, and the overall change is small.

From the perspective of interference degree, the interference degree of the natural forest landscape increased, and the large human activity and planting of rubber increased to the forest landscape. The water landscape decreased overall, and the disturbance degree of rubber and cultivated landscape decreased, especially at the later stage from 2003 to 2010. Therefore, the interference from construction land was significantly increased. In the later stages, urban construction became concentrated and formed on a larger scale.

### Space and temporal analysis and prediction

**Ecological security index calculation and prediction model.** The Xishuangbanna landscape was predicted by grey prediction model (1,1). The $R^2$ was selected for the root mean square error (RMSE). The RMSE represents the degree of discretization of the predicted values, with an RMSE of 0 for the best fit. The error test values are shown in Fig 4, The $R^2$ value close to 1 indicates that the prediction model is better, and this model can be used to predict future landscape ecological security changes. The RMSE value is close to 0, indicating good accuracy and a more reliable prediction value.

The semi-variation function model fitting obtains the semi-variation function of the four-phase ecological safety index (Table 2). According to the test calculation, the index models in 1996, 2003, and 2010 and other phase spherical models, the fitting effect is more ideal, the ratio of block gold value and base value was below 20%, and the correlation of variables is more significant. Therefore, the index model and spherical model are analyzed with spatial analysis. According to the ERI calculation formula, each risk community's landscape ecological safety index was obtained. According to relevant studies, the ArcGIS Natural Breaks classification method was divided into five grades by appropriate modification (Table 3). Based on this, with the help of the theoretical model of variation function, the ArcGIS software interpolated the ecological security index of 243 risk communities obtained the landscape ecological security level map of the study area (Fig 5), and the ecological security area and proportion of each level were calculated (Fig 6). In 2023 and 2030, the landscape ecological safety were predicted from the GM.

**Landscape ecological security analysis.** Combined with Fig 5 and the calculation results, the overall landscape ecological safety in the study area was good in 1996, with grade I and grade II areas accounting for 63.5% and grade V only accounting for 9.5%. Except for the high safety level, the other grades are distributed in the northeast of Menghai County and the northwest of Jinghong City. The rubber forests and tea gardens, towns, and cultivated land are concentrated in this area.

In 2003, the proportion of grade I and grade II areas decreased to 53.4%, and the grade V area accounted for 20.4%, up by 10.9% compared with that of 1996. The scope spread from the west of Menghai County to Jinghong City, and the landscape ecological safety problems in the research area began to appear. The grade I area is mainly concentrated in the eastern part of Mengla County. The class V area spreads in Menghai County and Jinghong City, but predominantly in the central part, which accounts for the north part of Menghai County and the west part of Jinghong City. In 2010, the overall landscape ecological security reached 74.5%, which was the largest proportion of more than 20 years. The grade V area accounts for only 9% and is mainly distributed along the border of Menghai County and central Jinghong City. In 2017, the area of grade V area increased further, accounting for 35.8%. From the south of Jinghong, low safety areas also appeared in eastern Mengla County because the rubber was planted in the south of Jinghong, destroying the original natural forest. Furthermore, the class I area accounted for 24% and was concentrated in the north of Mengla County and the northeast of Jinghong City.

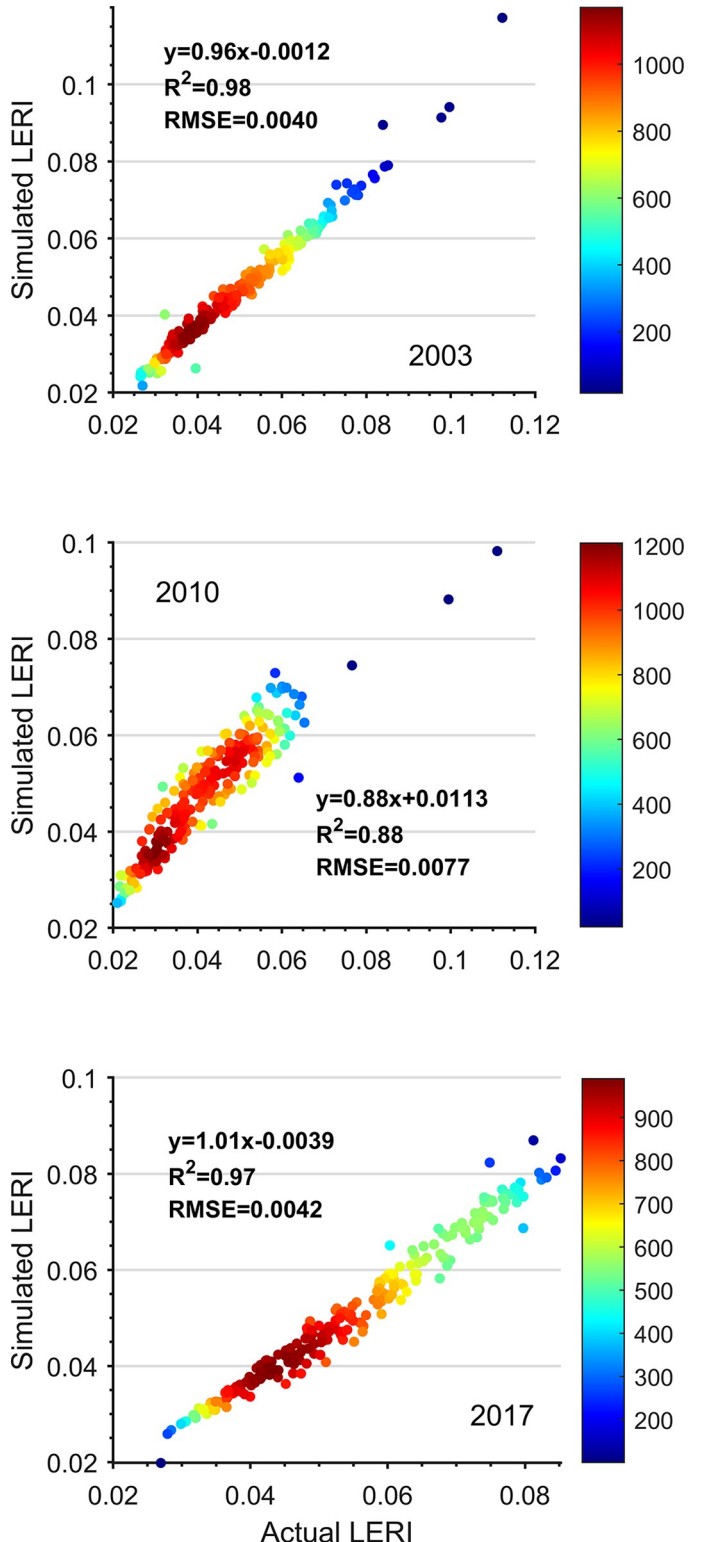

**Fig 4. Prediction accuracy of the Xishuangbanna landscape ecological safety grey model.**

**Table 2. Relevant parameters of the variant function model.**

| Year | Fitting model | Nugget $C_0$ | Sill $C + C_0$ | $C_0/(C + C_0)$ | Range $A/(m)$ |
|------|---------------|--------------|----------------|-----------------|----------------|
| 1996 | exponential model | $0.90281 \times 10^{-5}$ | $0.48700 \times 10^{-4}$ | 18.5% | 39951.9 |
| 2003 | exponential model | $0.30146 \times 10^{-4}$ | $0.14512 \times 10^{-3}$ | 20.75% | 32461.2 |
| 2010 | exponential model | $0.24545 \times 10^{-4}$ | $0.17858 \times 10^{-3}$ | 13.7% | 60944.7 |
| 2017 | spherical model | $0.83064 \times 10^{-3}$ | $0.50152 \times 10^{-2}$ | 16.6% | 234307.5 |
| 2023 | Spherical, model | $0.84074 \times 10^{-4}$ | $0.654925 \times 10^{-3}$ | 12.9% | 0.868635 |
| 2030 | spherical model | $0.212721 \times 10^{-3}$ | $0.148407 \times 10^{-2}$ | 14.3% | 0.8862052 |

According to the forecast, the situation eased in 2023, with grade I and grade II regions increasing compared to 2017. However, the proportion of grade V increased, and the transformation from grade IV to grade V increased rapidly. In 2030, the proportion of grade I regions increased further to 43.3%, accounting for 39.94% and reaching the highest value of more than 20 years. grade IV and grade V areas are similar to those in 2017, mainly concentrated in the southern parts of Jinghong and the south and southwest of Mengla. According to the ecological security distribution of the Xishuangbanna landscape from 1996 to 2030, it can be seen that there are still serious ecological security problems in places with high forest coverage rates and good preservation of the original tropical rainforests. This also demonstrates that the invisible ecological security problems cannot be ignored when a single economic forest replaces the original forest.

**Analysis of landscape ecological security transfer.** Fig 7 shows the Sankey diagram of changes in landscape ecological security levels in Xishuangbanna from 1996 to 2030. During 1996–2010, the overall ecological security showed a declining trend, with a significant transfer of a large number of Level I areas to lower ecological security levels. Specifically, in 2003, the areas classified as grade I increased their transfer to grade II, III, IV, and V areas by 2410.59 km$^2$, 949.99 km$^2$, 31.21 km$^2$, and 1.23 km$^2$, respectively. However, the grade V increased by 15.57% in 2010 compared to 2003, indicating a more severe ecological risk. From 2010 to 2017, the ecological security mainly shifted to higher levels. In this period, the transfers from grade II, III, IV, and V areas to grade I area increased by 3222.36 km$^2$, 2172.24 km$^2$, 518.04 km$^2$, and 1499.04 km$^2$, respectively. According to the predictions, in 2023–2030, there will be relatively minor changes in the ecological security levels, but a trend of polarization will emerge. Analyzing the changes in each ecological security level, it is evident that grade II and

**Table 3. Landscape ecological safety level in the study area.**

| Ecological risk level | Range | Status | Characteristics |
|-----------------------|-------|--------|-----------------|
| I | 0~0.042 | High safety | The ecosystem is rarely disturbed or damaged. The ecosystem structure is perfect. It is rich in tree species structure, presents strong recovery ability, and ecological problems are not significant. It is in a state of harmonious development between man and nature. |
| II | 0.042~0.049 | High safety | The ecosystem is relatively perfect, with less damage and good function, and can be automatically restored after interference. |
| III | 0.049~0.056 | Medium safety | The ecosystem service function has been degraded, and the ecological environment will be damaged to a certain extent. The ecosystem structure changes, but it can maintain basic functions. It is easy for it to deteriorate after receiving interference, and ecological problems will occur. |
| IV | 0.056~0.06 | Low safety | The ecosystem is greatly disturbed and damaged by people and has incomplete functions. The ecosystem structure is more likely to deteriorate, and ecological problems are large. |
| V | 0.06~1 | Low safety | The ecosystem is greatly disturbed and damaged, the ecosystem structure is single, it presents weakened function, and ecosystem problems are more prominent. |

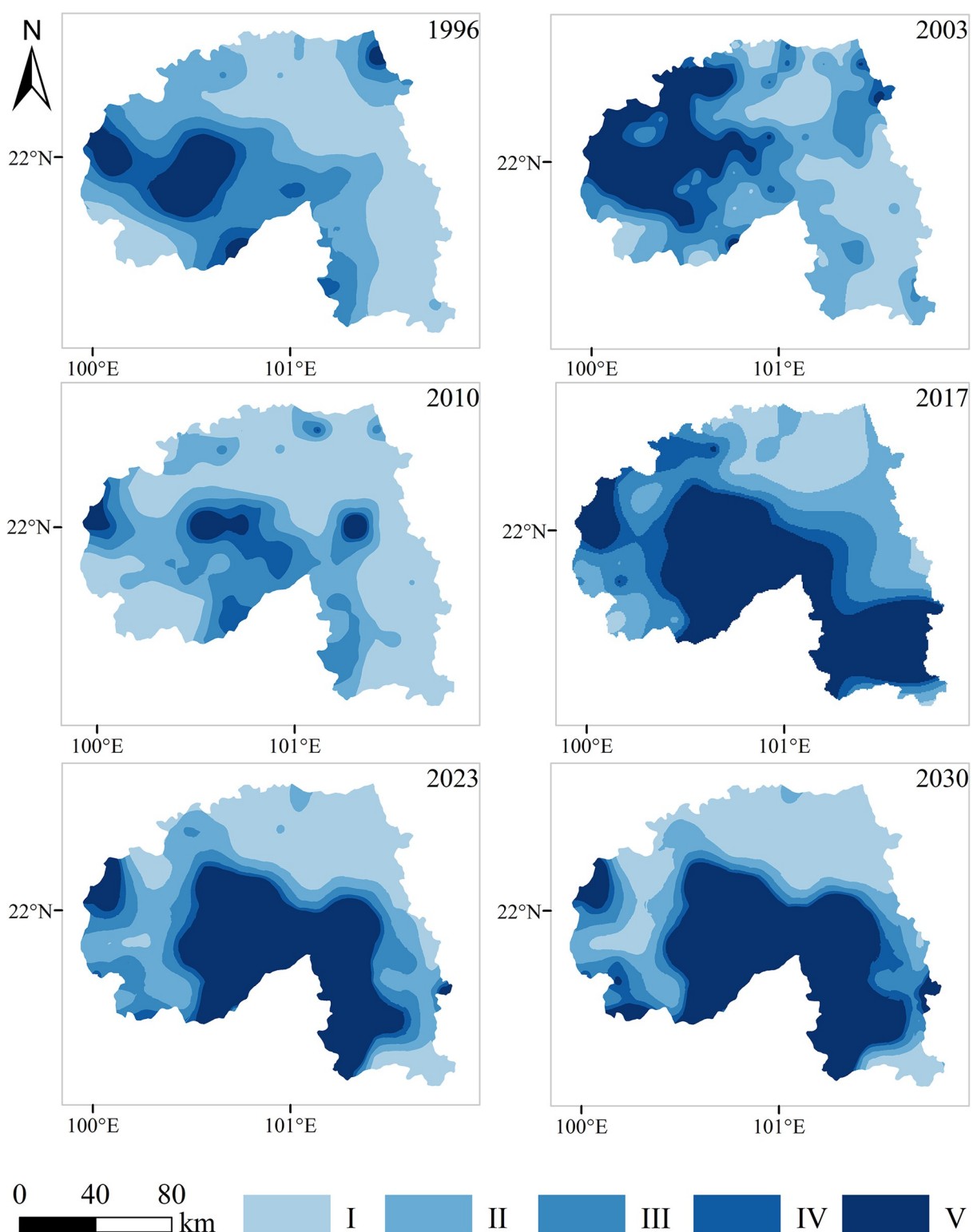

**Fig 5. Landscape ecological safety classification map of the study area from 1996 to 2030.**

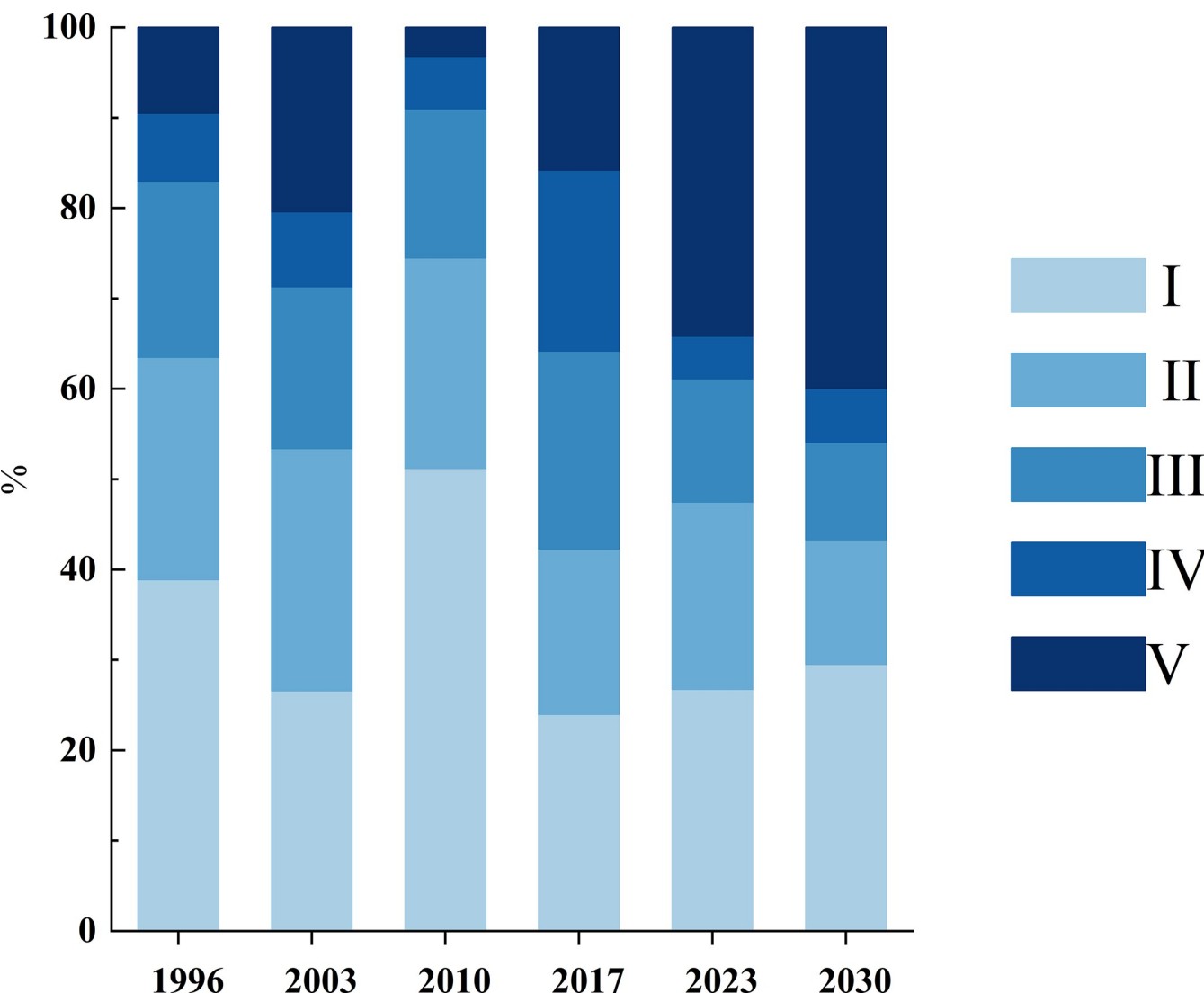

**Fig 6. Scale map of the landscape ecological safety level in the study area.**

III areas decreased annually, while the grade V area increased from 0.09% in 1996 to 39.9% in 2030. The proportion of grade I areas also declined from 38.9% to 29.6%.

## Spatial autocorrelation of landscape ecological security

The *Moran's I* were 0.8765, 0.8676, 0.789, 0.859, 0.863 and 0.871 in 1996, 2003, 2010, 2017, 2023 and 2030 (Table 4). The scores of the ecological risk value Z in different years were all greater than 1.65. This indicates that the ecological security of the Xishuangbanna landscape has obvious positive spatial correlation during the study period. The study area is basically a high value aggregation area and a low value aggregation area, and the area of high and low value aggregation regions is relatively small.

Further analysis was conducted on the local spatial correlation of landscape ecological security in Xishuangbanna, and local spatial autocorrelation cluster maps were obtained. The spatial distribution of the landscape ecological security index in most grid cells in the study area

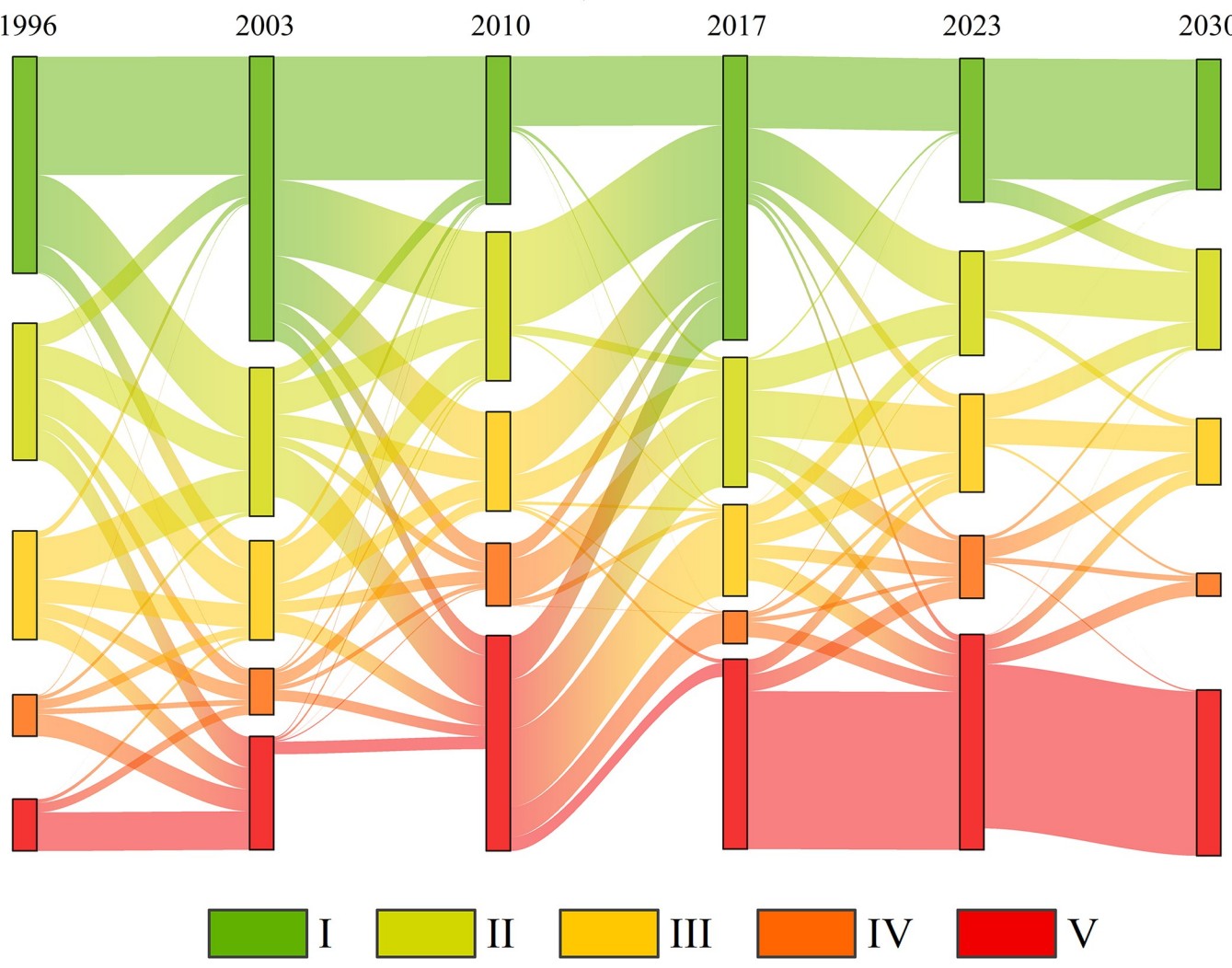

**Fig 7. Sankey diagram depicting landscape ecological security transformation, 1996–2030.**

from 1999 to 2030 exhibited a "high-high" (HH) and "low-low" (LL) aggregation pattern (Fig 8). However, there was a trend of a southeastward shift in the aggregation of low-value areas. The HH aggregation areas in the six time periods exhibited a relatively concentrated distribution in the tropical primary forests of the eastern Mengla, northern Jinghong, and southwestern Menghai, characterized by rugged terrain, low human disturbance, high landscape dominance, and lower fragmentation compared to other areas, resulting in high landscape ecological security index values. The distribution of LL aggregation areas expanded gradually southward from Menghai, mainly concentrated in the southern parts of Jinghong and Mengla.

**Table 4. The global *Moran's I* of landscape ecological patterns value and its test.**

| Year | 1996 | 2003 | 2010 | 2017 | 2023 | 2030 |
|---|---|---|---|---|---|---|
| global *Moran's I* | 0.8765 | 0.867 | 0.789 | 0.859 | 0.863 | 0.871 |
| P value | 0.001 | 0.001 | 0.001 | 0.001 | 0.001 | 0.001 |
| Z value | 18.7372 | 18.5572 | 17.0183 | 18.0308 | 18.1608 | 18.0746 |

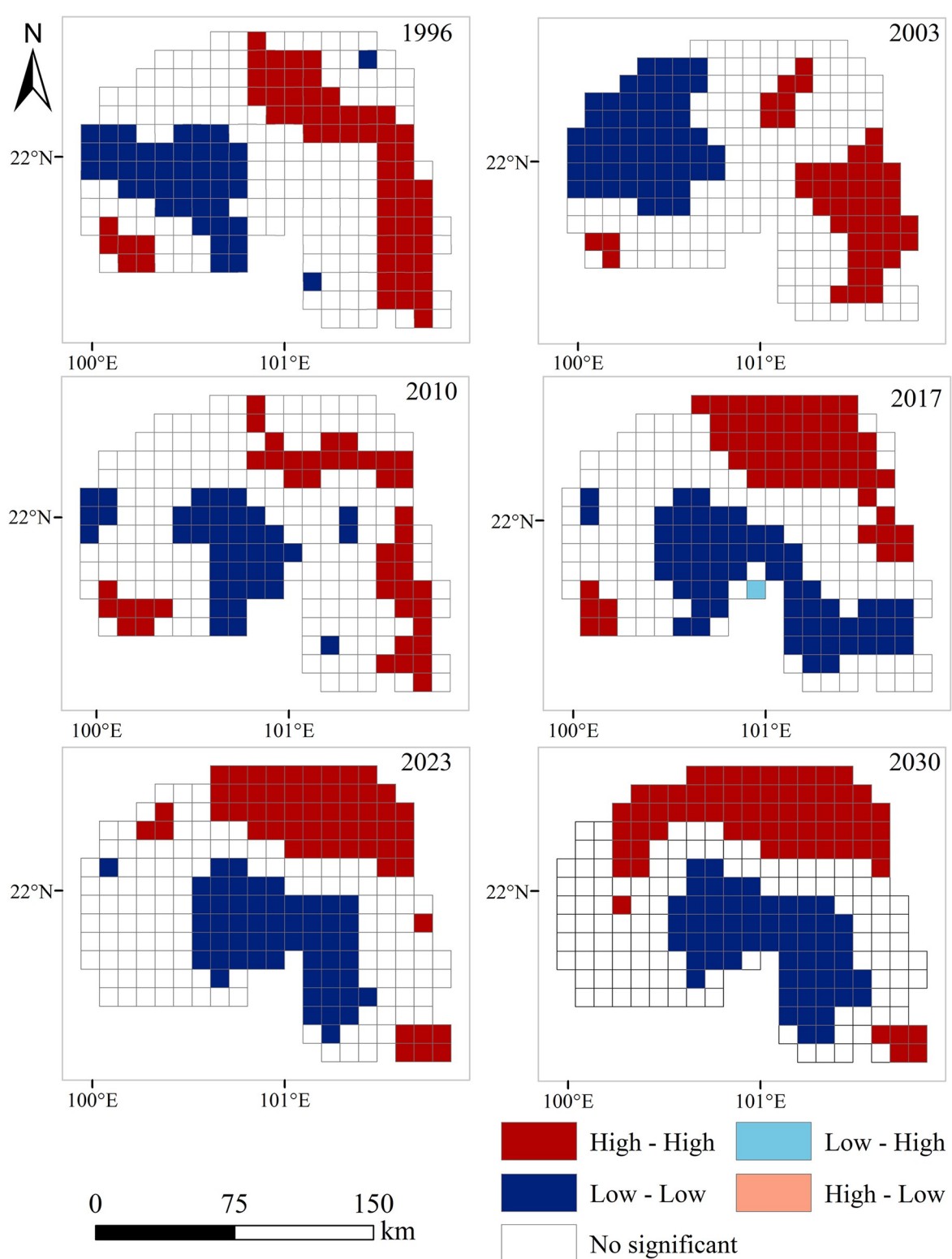

**Fig 8. Spatial clustering characteristics of landscape ecological security.**

This is attributed to increased human disturbance, expansion of rubber plantations, and extensive replacement of pristine forests with monoculture economic plantations, resulting in low landscape ecological security values. Xishuangbanna had only a few scattered instances of "low-high" (LH) and "high-low" (HL) aggregation areas, indicating that the landscape ecology in the study area remained relatively stable and less prone to rapid changes. According to the predictions, the spatial distribution and aggregation of HH areas will continue to expand eastward, while LL areas will extend further south, leading to intensified aggregation from 2023 to 2030.

### Analysis of the center of gravity change and shift

The center of gravity transfer map (Fig 9) generated from the coordinates of the center of gravity of the four phases of landscape ecological security grades can be used to obtain its center of gravity transfer characteristics (Table 5). It can be seen that the Xishuangbanna landscape ecological security space dynamic evolution characteristics from 1996 to 2030 were significant, showing the overall movement trend of "east to south" and eventually shifted to the southeast.

The center of gravity transfer in class I has shifted from the northeast to the northwest. The movement angles are 77.6˚, 104.3˚, 153.2˚, 32.8˚, and -83.5˚. The transfer distances were 118.45 km, 41.52 km, 2.08 km, 37.55 km, and 14.62 km, respectively. The level I region expands over time from the central town to the periphery. The center of gravity of the primary area shifts first to the southwest and then to the southeast, where the moving distance is not as large as in the level I area. The center of gravity of the level area is gradually close to the town. The center of gravity shifts first to the southeast, then to the southwest, and finally to the northwest. The center of gravity of the primary area is shifted from east to west. The center of gravity of the grade V area gradually becomes closer to the large area of planted rubber and rapid urbanization. The safety center of gravity is shifted from the southeast to the northwest.

## Discussion

### Ecological security index and ecological security

Forest degradation has been a serious problem in and around Xishuangbanna [50, 51]. Forest cover in Xishuangbanna has decreased from 69% to less than 50%, especially from 2003–2010, with an average annual decrease of 35,137 ha [52],and the important tropical seasonal rainforest landscape has decreased from 10.9% to 3.6% [53]. The landscape pattern has shifted from historically dominant natural tropical forests to large areas of planted rubber woodlands and tea plantations, with policy being one of the main drivers behind the expansion of rubber forests and tea plantations [54], and beginning in the 1990s, the government began to encourage rubber plantations, and large areas of rainforests have been converted to plantations, replacing natural and secondary forests [51, 55]. Deforestation, overexploitation, and forest fragmentation have challenged the fragility of the furniture ecosystem and put the tropical forest ecosystem under greater ecological pressure [56]. Therefore, rubber forests in forest land were proposed to be classified into one category during land use classification as a way to analyze and rate the fragmentation and landscape ecological security rating of each landscape in Xishuangbanna.

Our results are consistent with previous findings that rubber plantations are rapidly expanding in Xishuangbanna [57], with the rubber forest area increasing by $31.3 \times 10^4 hm^2$ from 1999 to 2010, which is a one-fold increase in area compared to 1996. The distribution of rubber forests highly overlapped with the distribution of low-level landscape ecological security, and most of the rubber forests and their surrounding areas were class IV and V areas. The large expansion of monoculture plantations has reduced the degree of landscape separateness

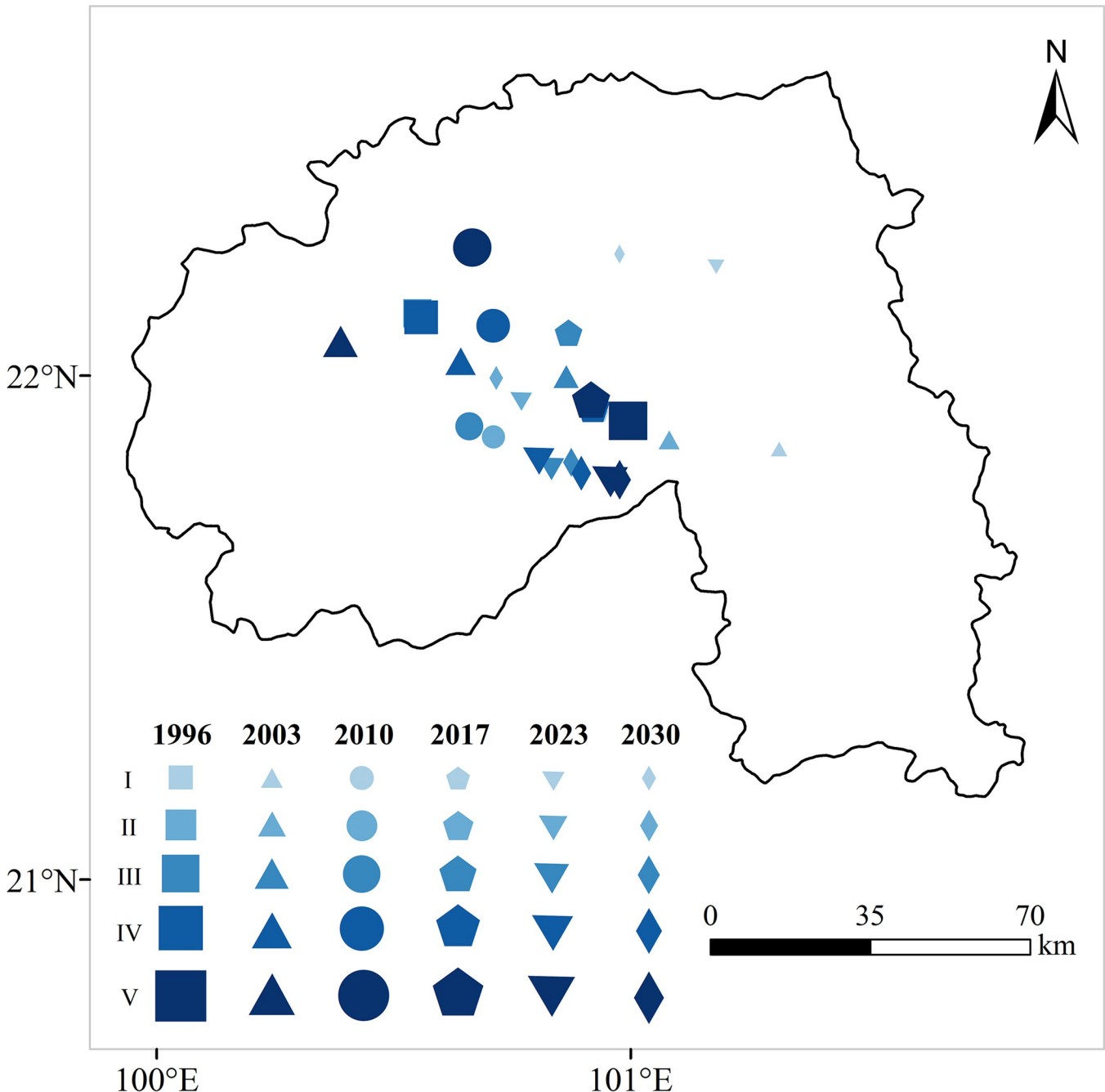

**Fig 9. Center of gravity shift of landscape ecological security level type in Xishuangbanna from 1996 to 2030.**

but increased disturbance, which is expected to reduce carbon stocks [55] and negatively affect biodiversity and connectivity [58], with implications for landscape ecological security.

## Tempo-spatial changes of landscape ecological security

Since the introduction of rubber to Xishuangbanna, rubber plantations have rapidly expanded into many low-hill areas with the support of national technology, replacing most of the native vegetation [59]. Overall, the landscape ecological security of Xishuangbanna in 2017 showed

**Table 5. Statistical distance and angle of landscape ecological security level in Xishuangbanna from 1996 to 2016.**

| Landscape ecological security level | Time | Variation of longitude (km) | Variation of latitude (km) | Mobile Angle (°) | Move direction | Moving distance (km) |
|---|---|---|---|---|---|---|
| I | 1996~2003 | -115.49 | -26.29 | 77.6 | northeast | 118.45 |
| | 2003~2010 | 40.38 | -9.67 | 104.3 | northwest | 41.52 |
| | 2010~2017 | -0.96 | 1.84 | 153.2 | northwest | 2.08 |
| | 2017~2023 | -20.02 | -31.77 | 32.8 | east, north | 37.55 |
| | 2023–2030 | 14.54 | -1.44 | -83.50 | southeast | 14.62 |
| II | 1996~2003 | 8.79 | 8.52 | 226.5 | southwest | 12.24 |
| | 2003~2010 | 38.60 | -0.6949 | 91.7 | northwest | 38.6 |
| | 2010~2017 | -22.36 | -7.17342 | 72.8 | northeast | 23.49 |
| | 2017~2023 | 17.47 | 0.12 | -89.70 | southeast | 17.47 |
| | 2023~2030 | 4.70 | -3.50 | -52.50 | southeast | 5.86 |
| III | 1996~2003 | 10.55 | -3.63 | -70.1 | southeast | 11.16 |
| | 2003~2010 | 21.28 | 10.91 | 243.4 | southwest | 23.92 |
| | 2010~2017 | -21.84 | -20.50 | 53.6 | northeast | 29.96 |
| | 2017~2023 | 7.91 | 28.54 | 196.10 | southwest | 29.61 |
| | 2023~2030 | -5.09 | 0.03 | 91.00 | northwest | 5.09 |
| IV | 1996~2003 | -8.66 | 9.71 | -20.8 | southeast | 13.01 |
| | 2003~2010 | -7.21 | -7.74 | 43.4 | northeast | 10.58 |
| | 2010~2017 | 21.91 | 17.69 | 129.7 | northwest | 28.16 |
| | 2017~2023 | 14.62 | 10.47 | 235.00 | southwest | 17.98 |
| | 2023~2030 | -10.51 | 5.47 | 118.30 | northwest | 11.85 |
| V | 1996~2003 | 7.77 | -12.19 | -68.2 | southeast | 14.45 |
| | 2003~2010 | -28.96 | -20.65 | 54.9 | northeast | 35.57 |
| | 2010~2017 | -26.16 | 33.39 | 142.7 | northwest | 42.42 |
| | 2017~2023 | -2.47 | 17.69 | 172.80 | northwest | 17.86 |
| | 2023~2030 | -4.24 | 1.71 | 112.90 | northwest | 4.57 |

some problems, with only 32.32% of Class I and II areas, located in the northern part of Jinghong and Mengla, with less anthropogenic interference due to the topography and protection policies. The percentage of Class IV and V areas reached 50.05%, mostly located in the southern part of Xishuangbanna's rubber forest plantations.

The landscape ecological security of Xishuangbanna can be divided into three phases: 1996–2003, 2003–2010, and 2010 onwards. During 1996–2003, the proportion of class I areas decreased from 38.98% to 26.61%, and the proportion of class V areas increased from 9.34% to 20.4%. Starting from the 1990s, human beings have extensively exploited natural resources without considering environmental issues, and ecological risks have continued to increase [60], and Jinghong City represents a decline in ecological security, with a large population flow and stronger negative effects from human disturbance activities [61]; during the period of 2003–2010, the percentage of class I areas increased from 26.59% to 51.14%, while the percentage of class V areas decreased from 20.38% to 3.17%. In the 2000s, human beings continued to pursue economic benefits, but gradually realized the importance of environmental protection, and the ecological quality was improved to a certain extent [62]; after 2010, the proportion of class V areas sharply decreased from 51.14% to 13.56%. Rubber plantation areas in Xishuangbanna have shown a clear expansion trend from concentrated to decentralized, with continued proliferation in the Sino-Lao (near Luang Namtha) and Sino-Myanmar border areas (Shan State border areas), and increased fragmentation of the regional habitats [63, 64].

## Contributions and limitations

Xishuangbanna is located in the southwest border area of China, and its ecological security problem is a typical cross-border ecological problem. Through the construction of landscape loss degree and other indexes, the landscape risk assessment model evaluates the landscape ecological safety of Xishuangbanna and analyzes the characteristics of spatial and temporal division using the center of gravity transfer, which can better reflect the spatial and temporal changes of the ecological security landscape in the study area [12]. However, some key ecological processes that play an important role in the ecological security of the study area are often underexplained. Firstly, the study highly relies on the results of land use classification, in which the overall accuracy of land use classification in 1996, 2003, 2010 and 2030 was higher than 85%, which is a reliable basis for assessing the ecological security of the landscape. However, errors in classification are inevitable, so improving the accuracy of land use data is an important direction for future research. Second, ecological restoration can alleviate ecological degradation and improve the ecological security grade [65], but compared to natural forests planted forests may affect the grade of landscape ecological security, and as ecological restoration continues to advance, the comparative evaluation of planted forests and natural forests should also be the focus of future regional research [66]. Thirdly, spatial and temporal changes in landscape ecological security, such as human activities and climate change, should be addressed in the future [7, 67].

In the early days of policy and economic interests, large areas of rubber forests and other economic forests existed. At present, with the global development of artificial synthetic rubber technology, the trend of replacing natural rubber is obvious, which has also led to a decline in the price of natural rubber. As a hidden ecological risk, rubber forests need to be transformed under policy guidance to avoid causing cross-border ecological security problems.

## Conclusion

This study is based on land use classification. With the help of "3S" technology, landscape ecology theory, the gray prediction model, temporal evolution characteristics, and the development trends of landscape ecological security in Xishuangbanna from 1996 to 2030, the following research results have been obtained.

1. In more than 20 years, the forest land landscape marked by the tropical rain forest has been greatly reduced, and the forest land has decreased by $40.810^4 \, hm^2$. The originally complete tropical rainforest fields are fragmented, and the landscape fragmentation throughout the whole tropical forest vegetation increases significantly. It is accompanied by the rapid increase in rubber forest and tea plantation area becoming a large artificial vegetation landscape, gradually replacing the original natural tropical forest. The water landscape area decreased, forming small and fragmented landscape patches, and the degree of landscape fragmentation increased.

2. The ecological security in Xishuangbanna has experienced fluctuations. In 1996, the areas designated as Level I and II ecological safety constituted 63.5% of the total land, but this percentage decreased to 53.4% in 2003. By 2010, this proportion had risen to 74.5%, which was the highest in over two decades. Conversely, the areas under Level IV and V increased by 2017, which has intensified concerns about ecological safety. Our projections for 2023 and 2030 indicate that the proportion of Level I and II areas will further increase to 43.3%, while Level V areas will reach an all-time high of 39.94%. These changes will primarily occur in the southern regions of Jinghong and Mengla.

3. The regional difference in the change distance of the landscape center of gravity of different landscape ecological safety levels is noticeable. Spatial aggregation has strengthened annually, with High-High and Low-Low clusters becoming more concentrated. Due to human activities, the spatiotemporal characteristics of landscape ecological security in Xishuangbanna have been significantly altered from 1996 to 2030, generally moving in a "East-South" directional trend and eventually shifting southeastward.

Although the Xishuangbanna forest coverage rate is high, it is mainly covered by the rubber forests. Some abandoned farmlands gradually restored secondary forest increase. Furthermore, the original rainforest area's high ecological service function, it is significantly reduced and replaced by the low ecological service of rubber forests and tea gardens. In other words, serious ecological security problems are hidden under the background of high forest coverage. Therefore, after the original tropical rainforest is replaced by a single economic forest, the invisible ecological security problem of Xishuangbanna cannot be ignored.

## Acknowledgments

Our special thanks go to Prof Yang Yuming for providing constructive suggestions.

## Author Contributions

**Conceptualization:** Zhuoya Zhang, Hailong Ge, Xiaona Li.

**Data curation:** Xiaoyuan Huang.

**Formal analysis:** Xiaoyuan Huang.

**Methodology:** Zhuoya Zhang, Siling Ma, Qinfei Bai.

**Resources:** Zhuoya Zhang.

**Software:** Hailong Ge.

**Visualization:** Xiaona Li.

**Writing – original draft:** Hailong Ge.

**Writing – review & editing:** Zhuoya Zhang, Xiaona Li, Xiaoyuan Huang.

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
