## [Decision Letter · Decision Letter 0]

13 Jul 2023

PONE-D-23-15562Spatiotemporal evolution and trend prediction of the Xishuangbanna landscape ecological security pattern from 1996-2030PLOS ONE

Dear Dr. Huang,

Thank you for submitting your manuscript to PLOS ONE. After careful consideration, we feel that it has merit but does not fully meet PLOS ONE’s publication criteria as it currently stands. Therefore, we invite you to submit a revised version of the manuscript that addresses the points raised during the review process. Please submit your revised manuscript by Aug 27 2023 11:59PM. If you will need more time than this to complete your revisions, please reply to this message or contact the journal office at plosone@plos.org. Please include the following items when submitting your revised manuscript:A rebuttal letter that responds to each point raised by the academic editor and reviewer(s). You should upload this letter as a separate file labeled 'Response to Reviewers'.A marked-up copy of your manuscript that highlights changes made to the original version. You should upload this as a separate file labeled 'Revised Manuscript with Track Changes'.An unmarked version of your revised paper without tracked changes. You should upload this as a separate file labeled 'Manuscript'.

We look forward to receiving your revised manuscript.

Kind regards,

Salim Heddam

Academic Editor

PLOS ONE

Journal Requirements:

"This research was funded by the National Natural Science Foundation of China, grant number 42171240 and 31901322; Research start-up fund from Southwest Forestry University, grant number112125; Yunnan Dianchi Lake Protection and Management Foundation, Biodiversity Survey in Dianchi Lake Basin, grant number 2063010."

5. We note that Figures 1,2 4 & 6 in your submission contain map images which may be copyrighted. All PLOS content is published under the Creative Commons Attribution License (CC BY 4.0), which means that the manuscript, images, and Supporting Information files will be freely available online, and any third party is permitted to access, download, copy, distribute, and use these materials in any way, even commercially, with proper attribution. For these reasons, we cannot publish previously copyrighted maps or satellite images created using proprietary data, such as Google software (Google Maps, Street View, and Earth). For more information, see our copyright guidelines: http://journals.plos.org/plosone/s/licenses-and-copyright.

(1) You may seek permission from the original copyright holder of Figures 1,2 4 & 6 to publish the content specifically under the CC BY 4.0 license.  

**Additional Editor Comments:**

Reviewer 1:In this study, ‘transboundary eco-security’ was selected as the entry point for the study.

Based on the change of land use in Xishuangbanna, with the help of "3S" technology, landscape ecology theory, and gray prediction model, the spatial and developmental trends of landscape ecological security in Xishuangbanna from 1996-2030 could be determined.

The language of the whole essay is fluent, and the selected topic has certain research significance, but there are many places where the elaboration is unclear and insufficient to explore.

1）In the section of 2.3.2, why are 8km/10km and 12km chosen as the study area evaluation scale for comparison? Why do terrain and slope serve as the basis for evaluation units in the study area? How was the comparison made?

2) In the results section, the current findings can be further analyzed. For example, is there a spatial and temporal correlation of ecological safety patterns? How does the distribution of landscape ecological safety patterns vary from year to year for different grades?

3) The discussion section should be further discussed in depth. It should corroborate the feasibility and accuracy of the results of the research through previous studies, rather than just stating the importance of this study. In addition, the discussion in this paper is not deep enough and should be expanded according to the contents.

Reviewer 2:In this paper, the author developed the landscape ecological risk index of Xishuangbanna from 1996 to 2017 to assess the landscape ecological security, and to predict the evolution trend of landscape ecological security from 2023 to 2030. This study can analysis the land cover/ land use change of Xishuangbanna based on ENVI and ArcGIS software, and a great deal of work has been done in evaluating and predicting the ecological security of landscapes. But there are many issues should be revised to improve clarity and fluency of the manuscript.

There are some comments.

Comment 1: The title of this article does not reflect the research contents well.

Comment 2: The English writing need improved for the requirement of the scientific paper.

Comment 3: The introduction has no smooth transition between paragraphs, lacks a description of the current state of research in the relevant field, and has a high rate of rewriting.

Comment 4: The location map of the study area should be added in the overview of the study area section.

Comment 5: The error of formula in research method section should be corrected, such as formula 3.

Comment 6: Line 153 to line 157, what is the number of the “68 rubber forests”? And the title of picture 1 should be revised.

Comment 7: The size of assessment grid should be written as 8 km*8 km, and the related selection criteria you referred should be cited.

Comment 8: In general, the description of the results is not clear enough, and the structure of this paper is unclear, such as section of 3.2 and 3.3 should be merged.

Comment 9: In the discussion part, the discussion is not profound enough, and the analysis of the results lacks references to support the conclusion. Conclusion should be refined.

Comment 10: The manuscript has more writing errors, which should be carefully checked by the author

Reviewers' comments:

Reviewer's Responses to Questions

**Comments to the Author**

1. Is the manuscript technically sound, and do the data support the conclusions?

Reviewer #1: Yes

Reviewer #2: Yes

2. Has the statistical analysis been performed appropriately and rigorously? 

Reviewer #1: Yes

Reviewer #2: Yes

3. Have the authors made all data underlying the findings in their manuscript fully available?

Reviewer #1: Yes

Reviewer #2: Yes

4. Is the manuscript presented in an intelligible fashion and written in standard English?

Reviewer #1: Yes

Reviewer #2: Yes

5. Review Comments to the Author

Reviewer #1: In this study, ‘transboundary eco-security’ was selected as the entry point for the study.

Based on the change of land use in Xishuangbanna, with the help of "3S" technology, landscape ecology theory, and gray prediction model, the spatial and developmental trends of landscape ecological security in Xishuangbanna from 1996-2030 could be determined.

The language of the whole essay is fluent, and the selected topic has certain research significance, but there are many places where the elaboration is unclear and insufficient to explore.

1）In the section of 2.3.2, why are 8km/10km and 12km chosen as the study area evaluation scale for comparison? Why do terrain and slope serve as the basis for evaluation units in the study area? How was the comparison made?

2) In the results section, the current findings can be further analyzed. For example, is there a spatial and temporal correlation of ecological safety patterns? How does the distribution of landscape ecological safety patterns vary from year to year for different grades?

3) The discussion section should be further discussed in depth. It should corroborate the feasibility and accuracy of the results of the research through previous studies, rather than just stating the importance of this study. In addition, the discussion in this paper is not deep enough and should be expanded according to the contents.

Reviewer #2: In this paper, the author developed the landscape ecological risk index of Xishuangbanna from 1996 to 2017 to assess the landscape ecological security, and to predict the evolution trend of landscape ecological security from 2023 to 2030. This study can analysis the land cover/ land use change of Xishuangbanna based on ENVI and ArcGIS software, and a great deal of work has been done in evaluating and predicting the ecological security of landscapes. But there are many issues should be revised to improve clarity and fluency of the manuscript.

There are some comments.

Comment 1: The title of this article does not reflect the research contents well.

Comment 2: The English writing need improved for the requirement of the scientific paper.

Comment 3: The introduction has no smooth transition between paragraphs, lacks a description of the current state of research in the relevant field, and has a high rate of rewriting.

Comment 4: The location map of the study area should be added in the overview of the study area section.

Comment 5: The error of formula in research method section should be corrected, such as formula 3.

Comment 6: Line 153 to line 157, what is the number of the “68 rubber forests”? And the title of picture 1 should be revised.

Comment 7: The size of assessment grid should be written as 8 km*8 km, and the related selection criteria you referred should be cited.

Comment 8: In general, the description of the results is not clear enough, and the structure of this paper is unclear, such as section of 3.2 and 3.3 should be merged.

Comment 9: In the discussion part, the discussion is not profound enough, and the analysis of the results lacks references to support the conclusion. Conclusion should be refined.

Comment 10: The manuscript has more writing errors, which should be carefully checked by the author

6. PLOS authors have the option to publish the peer review history of their article (what does this mean?). If published, this will include your full peer review and any attached files.

Reviewer #1: No

Reviewer #2: No

---

## [Author Response · Author response to Decision Letter 0]

7 Sep 2023

Dear Professors and editors,

Thank you very much for processing our manuscript (PONE-D-23-15562) entitled " Spatiotemporal evolution and trend prediction of the Xishuangbanna landscape ecological security pattern from 1996-2030", by Zhuoya Zhang, Hailong Ge, Xiaona Li, Xiaoyuan Huang, Siling Ma and Qinfei Bai. We highly appreciate your comments and revision suggestions. All the comments from the reviewers are constructive and valuable. We changed the title to "Spatiotemporal patterns and prediction of landscape ecological security in Xishuangbanna from 1996-2030" by integrating the reviewers' comments and suggestions. Our point-to-point responses and corrections are displayed with blue fonts in the following text. We have made major and careful revisions closely following the comments. Thus, the quality and clarity of our manuscript are improved. We sincerely hope our responses are satisfactory.

If you have any question, please let us know. 

Yours sincerely,

Xiaoyuan Huang

Associate Professor

Faculty of Geography and Ecotourism

Southwest Forestry University

Kunming, China, 650224

E-mail: hxy19792721@163.com

-Response to Reviewer 1 

Thank you very much for your time involved in reviewing the manuscript and your very encouraging comments on the merits.

1. In the section of 2.3.2, why are 8km/10km and 12km chosen as the study area evaluation scale for comparison? Why do terrain and slope serve as the basis for evaluation units in the study area? How was the comparison made?

Answer: Thank you for pointing out our problem. We apologize for not expressing our thoughts clearly. We acknowledge that there were issues with our expression. The selection of evaluation units in the study area is crucial for ecological safety evaluation. We have chosen the evaluation units based on the extent of the study area and the patch area of different landscape types. Additionally, we have appropriately cited the references used in this process. Specific modifications and improvements are as follows:

(1) Determination of the evaluation unit of the study area

Through the Fishnet Analysis tool in ArcGIS 10.8, taking into account the study area’s extent and the patch areas of different landscape types, following the 2-5 times principle based on patch size[42-44], the study area is divided into 243 square grid cells of 10×10 km each using an equidistant method. A risk index is then constructed to determine the risk value at the center point of each landscape unit, which is utilized to calculate the landscape ecological security index (Fig. 3).

2. In the results section, the current findings can be further analyzed. For example, is there a spatial and temporal correlation of ecological safety patterns? How does the distribution of landscape ecological safety patterns vary from year to year for different grades?

Answer: We would like to express our gratitude to the reviewers for this comment, and in the results section, we have included a spatial autocorrelation analysis of ecological security in Xishuangbanna landscapes and a transfer analysis of ecological security. The spatial autocorrelation analysis enables a comprehensive examination of the spatial correlation of ecological security patterns. Additionally, the transfer analysis facilitates the study of inter-annual distributional changes in the ecological security patterns of landscapes at various levels. The new chapter titles are Spatial autocorrelation analysis, Analysis of landscape ecological security transfer and Spatial autocorrelation of landscape ecological security.

3. The discussion section should be further discussed in depth. It should corroborate the feasibility and accuracy of the results of the research through previous studies, rather than just stating the importance of this study. In addition, the discussion in this paper is not deep enough and should be expanded according to the contents.

Answer: We appreciate you pointing that out. We have revised the discussion section to cover the following topics: Ecological Security Index and ecological security, Tempo-spatial changes of landscape ecological security, Contributions and limitations. The feasibility and accuracy of our findings were demonstrated through comparison with other experiments. Furthermore, the causes of changes in landscape ecological security were further discussed.

We would like to express our gratitude for the time you have dedicated to reviewing our manuscript and providing us with this valuable opportunity for improvement. Your input has been immensely helpful, and we have taken it into careful consideration during the revision process. We sincerely hope that you will find the revised version to meet your expectations.

-Response to Reviewer 2 

We deeply appreciate the time you invested in reviewing our manuscript and the kind words of encouragement regarding its merits. Your thoughtful evaluation has been invaluable to us. Thank you sincerely for your dedication and support.

1. The title of this article does not reflect the research contents well.

Answer: Thank you very much for your valuable comments on our paper. We have reconsidered the title and revised it according to your guidance. The revised title is: "Spatiotemporal patterns and prediction of landscape ecological security in Xishuangbanna from 1996-2030"

2. The English writing need improved for the requirement of the scientific paper.

Answer: Thank you for your careful review and valuable comments. We have asked a professional touch-up agency to touch up the language of the article before submission, however, under your guidance, we found that part of the writing may not be expressed accurately. To ensure that the language of the article is of the best standard, we have carefully reviewed the whole article again and made further revisions to address the parts that may be unclear or prone to misinterpretation. We expect our revisions to be at the level of a scientific paper

3. The introduction has no smooth transition between paragraphs, lacks a description of the current state of research in the relevant field, and has a high rate of rewriting.

Answer: Thank you very much for your suggestions for revision. As per your request, I have revised the introduction to address the lack of flow between paragraphs and added a description of the current state of research in the relevant field. In addition, I have reduced the frequency of rewrites.

The problem of poor articulation has been resolved. I have ensured that paragraphs are more naturally connected to each other by incorporating transition words and sentences.

To address the lack of description of the current state of research in the relevant field, I have cited the latest research findings and published papers that provide cutting-edge advances in the field today.

Finally, I have also made an effort to minimize the frequency of rewrites in order to ensure coherence and consistency in the text.

4. The location map of the study area should be added in the overview of the study area section.

Answer: Thank you very much for your review comments. As per your suggestion, I have added a location map of the study area in Overview of the study area.

Fig. 1 location of the study area

The map images we use come from Environmental Sciences, Chinese Academy of Sciences (http://www.resdc.cn/)

5. The error of formula in research method section should be corrected, such as formula 3.

Answer: Thank you very much for your review comments. I sincerely apologize for my formula errors in the research methods section.We have carefully examined and corrected the formula errors in the Research Methods section, including fixing Equation 3. We will ensure that the correct mathematical notation and expressions are followed during the revision process, with proper validation and review. Here is our revised formula 3:

6. Line 153 to line 157, what is the number of the “68 rubber forests”? And the title of picture 1 should be revised.

Answer: Thank you very much for your review comments. We apologize for the ambiguity in our expression, we originally meant: 68 out of 305 random sample points for rubber forests, and we have revised lines 153-157 as follows:

In this study, a total of 305 GPS sampling points were distributed along the main roads in Xishuangbanna Prefecture. Among these points, there were 68 rubber plantation areas, 73 tea gardens, 85 forested areas, 23 developed lands, 16 water bodies, and 40 cultivated lands The total accuracy Kappa coefficients after phase 5 image classification were 85.9%, 86.7%, 89.9%, 93.5%, and 87.3%, respectively, which met the study needs.

Meanwhile, we have revised the title of picture 1, it is now:

Fig. 2 Land use type changes in Xishuangbanna from 1996 to 2017

7. The size of assessment grid should be written as 8 km*8 km, and the related selection criteria you referred should be cited.

Answer: Many thanks to the reviewer for your guidance. I apologize for the omission in the description of the dimensions of the evaluation grid and the associated selection criteria. Following your suggestion, I will explicitly state the size of the evaluation grid as 10 km × 10 km in the text, and this correction will more accurately reflect the actual size used in the study. Combining your and Reviewer #1's suggestions, we have rewritten this section, and we have selected the evaluation cells based on the extent of the study area and the size of the patches in different landscape types. Specific modifications and improvements are as follows:

(1) Determination of the evaluation unit of the study area

Through the Fishnet Analysis tool in ArcGIS 10.8, taking into account the study area’s extent and the patch areas of different landscape types, following the 2-5 times principle based on patch size[42-44], the study area is divided into 243 square grid cells of 10×10 km each using an equidistant method. A risk index is then constructed to determine the risk value at the center point of each landscape unit, which is utilized to calculate the landscape ecological security index (Fig. 3).

8. In general, the description of the results is not clear enough, and the structure of this paper is unclear, such as section of 3.2 and 3.3 should be merged.

Answer: Many thanks to the reviewers for reviewing our paper and providing feedback. We apologize for the lack of clarity in the description of the results and the structure of the paper.

We acknowledge the concerns raised by the reviewers regarding the clarity of the result descriptions and the structure of the paper. We will make additional revisions to the paper based on your suggestions to ensure that the descriptions of the results are clearer. First, we have merged sections 3.2 and 3.3, which is now titled Space and temporal analysis and prediction. Additionally, we have divided this section into different tertiary headings: Ecological Security Index Calculation and Prediction Model; Landscape ecological security analysis; Analysis of landscape ecological security Taking into account your suggestion and that of reviewer #1, we have also added spatial autocorrelation analysis of ecological security and transfer analysis to enhance the analysis of landscape ecological security.

9. In the discussion part, the discussion is not profound enough, and the analysis of the results lacks references to support the conclusion. Conclusion should be refined.

Answer: Thank you very much for pointing out our problems. Taking into account your suggestions with reviewer 1, we have restructured the discussion section into three parts: Ecological Security Index and ecological security; Tempo-spatial changes of landscape ecological security; Contributions and limitations. By comparing our findings with those of other experiments, we have demonstrated their feasibility and accuracy. In addition, the reasons for the changes in landscape ecological security are further explored. At the same time, we further refine the conclusions to ensure that they accurately reflect the findings and contributions of the study. We clearly articulate the significance and potential implications of the conclusions.

10. The manuscript has more writing errors, which should be carefully checked by the author

Answer: Thank you for the detailed review. We have carefully and thoroughly proofread the manuscript to correct all the grammar and typos.

We would like to take this opportunity to thank you for all your time involved and this great opportunity for us to improve the manuscript. We hope you will find this revised version satisfactory.

---

## [Decision Letter · Decision Letter 1]

24 Sep 2023

PONE-D-23-15562R1Spatiotemporal patterns and prediction of landscape ecological security in Xishuangbanna from 1996-2030PLOS ONE

Dear Dr. Huang,

Thank you for submitting your manuscript to PLOS ONE. After careful consideration, we feel that it has merit but does not fully meet PLOS ONE’s publication criteria as it currently stands. Therefore, we invite you to submit a revised version of the manuscript that addresses the points raised during the review process.

We look forward to receiving your revised manuscript.

Kind regards,

Salim Heddam

Academic Editor

PLOS ONE

Journal Requirements:

Additional Editor Comments:

Reviewer 1:The author has done an excellent revision. I need to give appreciation to the author for the hard work done to improve the writing of the manuscript according to the reviewers' suggestions. There are some minor issues that need to be corrected, after which the manuscript can be considered approved for publication.

1）The title of line 222 is "Spatial analysis method", which is an inaccurate subheading. The methods in the text, such as spatial autocorrelation analysis, landscape index and landscape security pattern, are also spatial analysis methods. Consideration could be given to replacing them with a specific method.

2）There is a formatting error on line 174

3) Coordinates are missing in Figures 2, 3, 5, and 7

Reviewer 2:The manuscript was improved strongly compared with the previous edition, most of the references are state-of-the-art and well cited. The figures of the article are beautiful, the data is detailed and representative. The manuscript is very well structured and the conclusions are logical. I recommend this manuscript may be acceptable in its present form.

Reviewers' comments:

Reviewer's Responses to Questions

**Comments to the Author**

1. If the authors have adequately addressed your comments raised in a previous round of review and you feel that this manuscript is now acceptable for publication, you may indicate that here to bypass the “Comments to the Author” section, enter your conflict of interest statement in the “Confidential to Editor” section, and submit your "Accept" recommendation.

Reviewer #1: All comments have been addressed

Reviewer #2: All comments have been addressed

2. Is the manuscript technically sound, and do the data support the conclusions?

Reviewer #1: Yes

Reviewer #2: Yes

3. Has the statistical analysis been performed appropriately and rigorously? 

Reviewer #1: Yes

Reviewer #2: Yes

4. Have the authors made all data underlying the findings in their manuscript fully available?

Reviewer #1: Yes

Reviewer #2: Yes

5. Is the manuscript presented in an intelligible fashion and written in standard English?

Reviewer #1: Yes

Reviewer #2: Yes

6. Review Comments to the Author

Reviewer #1: The author has done an excellent revision. I need to give appreciation to the author for the hard work done to improve the writing of the manuscript according to the reviewers' suggestions. There are some minor issues that need to be corrected, after which the manuscript can be considered approved for publication.

1）The title of line 222 is "Spatial analysis method", which is an inaccurate subheading. The methods in the text, such as spatial autocorrelation analysis, landscape index and landscape security pattern, are also spatial analysis methods. Consideration could be given to replacing them with a specific method.

2）There is a formatting error on line 174

3) Coordinates are missing in Figures 2, 3, 5, and 7

Reviewer #2: The manuscript was improved strongly compared with the previous edition, most of the references are state-of-the-art and well cited. The figures of the article are beautiful, the data is detailed and representative. The manuscript is very well structured and the conclusions are logical. I recommend this manuscript may be acceptable in its present form.

7. PLOS authors have the option to publish the peer review history of their article (what does this mean?). If published, this will include your full peer review and any attached files.

Reviewer #1: No

Reviewer #2: No

---

## [Author Response · Author response to Decision Letter 1]

26 Sep 2023

-Response to Reviewer 1 

Thank you very much for your time involved in reviewing the manuscript and your very encouraging comments on the merits.

1. The title of line 222 is "Spatial analysis method", which is an inaccurate subheading. The methods in the text, such as spatial autocorrelation analysis, landscape index and landscape security pattern, are also spatial analysis methods. Consideration could be given to replacing them with a specific method.

Answer: Thank you very much for providing the feedback and suggestions again. We have carefully considered your suggestions and have changed the subtitle to "Landscape Ecological Security Analysis" in the revised version.

2. There is a formatting error on line 174

Answer: Thank you for pointing out our error, we have fixed the formatting error on line 174 and checked the formatting again to make sure there is no error.

3. Coordinates are missing in Figures 2, 3, 5, and 7

Answer: We appreciate you pointing that out. We have added latitude and longitude coordinate information to the revision and ensured that it is clearly labeled in the chart title or legend.

Fig. 2 Land use type changes in Xishuangbanna from 1996 to 2017

Fig. 3 The ecological security community of the study area

Fig. 5 Landscape ecological safety classification map of the study area from 1996 to 2030

Fig. 8 Spatial clustering characteristics of landscape ecological security

Fig. 9 Center of gravity shift of landscape ecological security level type in Xishuangbanna from 1996 to 2030

-Response to Reviewer 2 

Thank you very much for your review and valuable feedback. I am very happy to hear that you think my manuscript has improved significantly in comparison to the previous version and your comments on various aspects of the paper. This is very encouraging for me.

Thank you again for your review and support. I look forward to your further guidance and suggestions.

---

## [Decision Letter · Decision Letter 2]

2 Oct 2023

Spatiotemporal patterns and prediction of landscape ecological security in Xishuangbanna from 1996-2030

PONE-D-23-15562R2

Dear Dr. Huang

We’re pleased to inform you that your manuscript has been judged scientifically suitable for publication and will be formally accepted for publication once it meets all outstanding technical requirements.

Kind regards,

Salim Heddam

Academic Editor

PLOS ONE

Additional Editor Comments (optional):

Reviewer 1:The authors put efforts to response to the reviewer's comment. the manuscript can be accepted for publication at Plos One. Best wishes

Reviewer 2:The manuscript was improved strongly compared with the previous edition, most of the references are state-of-the-art and well cited.

The figures of the article are beautiful, the data is detailed and representative.

The manuscript is very well structured and the conclusions are logical. I recommend this manuscript may be acceptable in its present form.

Reviewers' comments:

Reviewer's Responses to Questions

**Comments to the Author**

1. If the authors have adequately addressed your comments raised in a previous round of review and you feel that this manuscript is now acceptable for publication, you may indicate that here to bypass the “Comments to the Author” section, enter your conflict of interest statement in the “Confidential to Editor” section, and submit your "Accept" recommendation.

Reviewer #1: All comments have been addressed

2. Is the manuscript technically sound, and do the data support the conclusions?

Reviewer #1: Yes

3. Has the statistical analysis been performed appropriately and rigorously? 

Reviewer #1: Yes

4. Have the authors made all data underlying the findings in their manuscript fully available?

Reviewer #1: Yes

5. Is the manuscript presented in an intelligible fashion and written in standard English?

Reviewer #1: Yes

6. Review Comments to the Author

Reviewer #1: The authors put efforts to response to the reviewer's comment. the manuscript can be accepted for publication at Plos One. Best wishes

7. PLOS authors have the option to publish the peer review history of their article (what does this mean?). If published, this will include your full peer review and any attached files.

Reviewer #1: No

---

## [Editor Report · Acceptance letter]

20 Oct 2023

PONE-D-23-15562R2 

Spatiotemporal patterns and prediction of landscape ecological security in Xishuangbanna from 1996-2030 

Dear Dr. Huang:

I'm pleased to inform you that your manuscript has been deemed suitable for publication in PLOS ONE. Congratulations! Your manuscript is now with our production department. 

Kind regards, 

on behalf of

Dr. Salim Heddam 

Academic Editor

PLOS ONE